# REASONING LIMITATIONS OF MULTIMODAL LARGE LANGUAGE MODELS. A CASE STUDY OF BONGARD PROBLEMS

## ABSTRACT

Abstract visual reasoning (AVR) encompasses a suite of tasks whose solving requires the ability to discover common concepts underlying the set of pictures through an analogy-making process, similarly to solving the human IQ test problems. Bongard Problems (BPs), proposed in 1968, constitute one of the fundamental challenges in this domain. Despite multiple advances in artificial intelligence, the BP tasks remain unsolved, mainly due to their requirement to combine visual reasoning and verbal description. In this work, we pose a question whether multimodal large language models (MLLMs) inherently designed to combine vision and language are capable of tackling BPs. To this end, we propose a set of diverse MLLM-suited strategies to tackle BPs and test 4 popular proprietary MLLMs: GPT-4o, GPT-4 Turbo, Gemini 1.5 Pro, and Claude 3.5 Sonnet, and 4 publicly available open models: InternVL2-8B, LLaVA-1.6 Mistral-7B, Phi-3.5-Vision, and Pixtral 12B. The above MLLMs are compared on 3 BP datasets from the AVR literature: a set of original BP instances relying on synthetic, geometry-based images and two recent datasets based on real-world images, i.e., Bongard-HOI and Bongard-OpenWorld. Our experiments reveal significant limitations of the current MLLMs in solving BPs. In particular, the models struggle to solve the classical set of synthetic BPs representing abstract concepts, despite their visual simplicity. Though their performance improves for real-world concepts expressed in Bongard-HOI and Bongard-OpenWorld datasets, the models still have difficulty in utilizing new information to improve their predictions, as well as utilizing the dialog context window effectively. To better capture the reasons of this performance discrepancy between synthetic and real-world AVR domains, we propose Bongard-RWR, a new BP dataset composed of specifically-designed real-world images that translate concepts from hand-crafted synthetic matrices to the real world, and perform focused experiments with this new dataset. The results suggest that weak models' performance on classical BPs is not due to the domain specificity, but rather comes from their general AVR limitations.

## 1 INTRODUCTION

Analogy-making is a critical aspect of human cognition, tightly linked with fluid intelligence, the capacity to apply learned skills in novel settings (Lake et al., 2017). Several approaches have been proposed to build systems capable of making analogies. Notably, the structure-mapping theory explores methods for discovering structural correspondences between pre-existing object representations (Winston, 1982; Gentner, 1983; Carbonell, 1983; Falkenhainer et al., 1989; Holyoak & Thagard, 1989). However, these approaches often overlook the perceptual aspect, assuming object representations are already given. Chalmers et al. (1992) highlight that forming useful representations is an intricate challenge. In particular, perception is not merely a passive reception of sensory data, but rather an active interpretation influenced by prior knowledge. This process involves the detection of patterns, recognition of analogies, and abstraction of concepts. The resultant representations may vary significantly depending on the context, which underscores the importance of modeling perception and cognition jointly (Hofstadter, 1995).

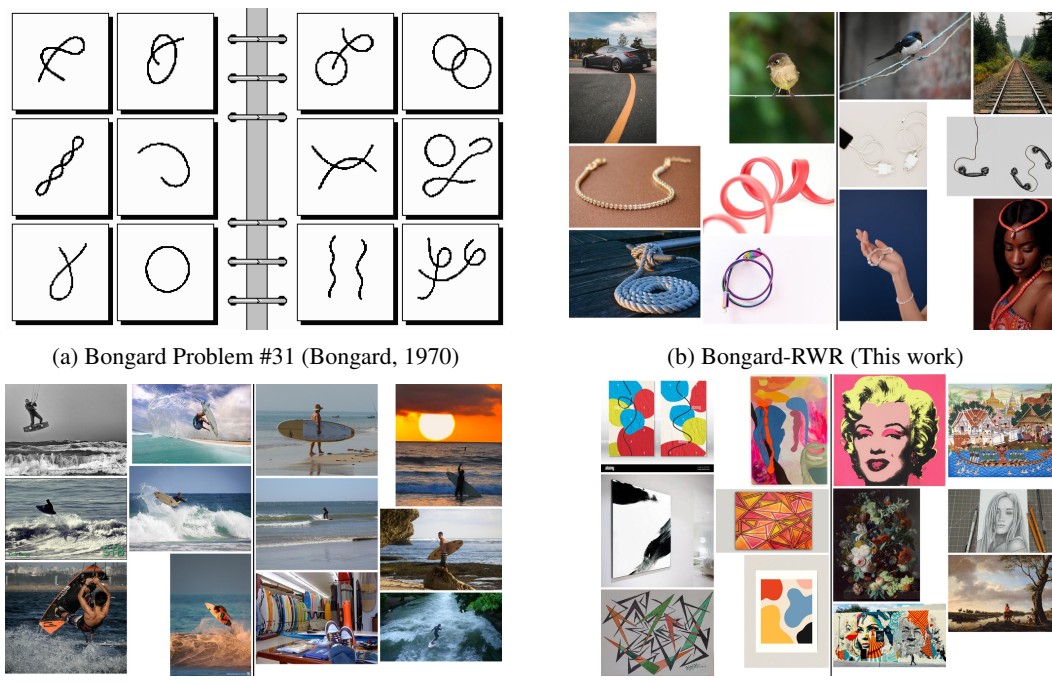

(a) Bongard Problem #31 (Bongard, 1970)    (b) Bongard-RWR (This work)

(c) Bongard HOI (Jiang et al., 2022)    (d) Bongard-OpenWorld (Wu et al., 2024)

Figure 1: **Bongard Problems.** In this work, we consider BPs with both synthetic and real-world images. (a) A manually designed BP #31. Left: One line. Right: Two lines. (b) A real-world representation of BP #31 from the proposed Bongard-RWR dataset introduced in Section 4. Left: One line. Right: Two lines. (c) Bongard HOI: Left: A person jumping on a surfboard. Right: Not a person jumping on a surfboard. (d) Bongard-OpenWorld: Left: An abstract painting. Right: Not an abstract painting. Descriptions of the Left / Right sides come from the respective datasets.

Multiple problems that necessitate combined perception and reasoning have been identified (Hofstadter, 1999). Among these tasks are Bongard Problems (BPs), introduced by Bongard (1968; 1970). Initial BPs were designed manually, leading to the formulation of a few hundred task instances by individual contributors (Foundalis, 2006b). A typical BP consists of two sides, left and right, each comprising six image panels arranged in a grid. All images on one side illustrate a shared concept absent in the images on the opposite side. The task is to identify the underlying rule that differentiates the sides and **articulate it in natural language**. Initial BPs (Bongard, 1968), akin to human IQ tests, featured abstract 2D geometric shapes, putting the focus on abstract reasoning. However, recent works have expanded the set of BPs to include real-world images, which broadens the scope of presented objects, attributes and relations. Specifically, the matrices in Bongard HOI (Jiang et al., 2022) depict human-object interactions, while Bongard-OpenWorld (Wu et al., 2024) employs open-world free-form concepts, increasing the diversity of featured scenes. Figs. 1a, 1c, 1d illustrate examples of problems from the three above-mentioned datasets.

A central theme in BPs is recognition of concepts in a context-dependent manner, as object representations need to be formed specifically for the presented matrix, rather than described *a priori* (Linhares, 2000). For example, consider the matrix in Fig. 1a – an analysis restricted to its left side may yield multiple concepts, such as the presence of curves or an object centered in the image. Only through a comprehensive understanding of both matrix sides one can recognize that the left side depicts a single line, while the right side presents two lines. Such concept-based tasks were argued to promote a more accurate evaluation of a system's generalization ability and its capacity for abstraction (Mitchell, 2021; Odouard & Mitchell, 2022). Moreover, the concepts in BPs are illustrated with several image examples, which positions the task within a few-shot learning setting (Fei-Fei et al., 2006; Wang et al., 2020). In contrast to other abstract reasoning problems, such as Raven's Progressive Matrices (RPMs) (Raven, 1936; Raven & Court, 1998; Małkiński & Mańdziuk, 2022) that have recently witnessed the development of large-scale benchmarks (Barrett et al., 2018; Zhang

et al., 2019), BPs allow to assess system's ability to derive concepts from a limited set of examples (typically six images per matrix side). The above aspects make BPs a valuable testbed for assessing abstract reasoning abilities of AI models.

**Motivation.** The quest to build systems capable of forming abstract concepts dates back to the 1950s (McCarthy et al., 2006). The advent of Deep Learning (DL) opened new possibilities to tackle BPs (Kharagorgiev, 2018; Nie et al., 2020). However, despite significant advancements, methods for consistently solving BPs (and other problems that involve abstract reasoning) are still lacking (Mitchell, 2021; van der Maas et al., 2021; Stabinger et al., 2021). Typically, DL approaches omit the generation of natural language answers by casting BP into a binary classification task, in which a test image had to be assigned to the matching side of the matrix. Conversely, a parallel stream of research on large language models (LLMs) demonstrated promising results in open-ended language generation (Brown et al., 2020). In particular, LLMs were applied to selected AVR tasks (Webb et al., 2023), though, lately Xu et al. (2024) pointed certain LLM limitations in solving AVR problems represented as text despite using information lossless translation through direct-grid encoding. Recent works have combined the vision and language modalities into multimodal large language models (MLLMs) (Achiam et al., 2023; Reid et al., 2024; Anthropic, 2024), inviting their application to diverse tasks (Yin et al., 2023; Wu et al., 2023). Motivated by these recent developments we examine the reasoning capabilities of MLLMs in solving BPs.

**Contributions.** The main contribution of this paper is four-fold.

(1) For the first time in the literature, we consider BPs in the context of MLLMs and propose a diverse set of strategies to solve BP instances in two setups: open-ended language generation and binary classification.

(2) We evaluate 4 state-of-the-art proprietary MLLMs and 4 open MLLMs on both synthetic and real-world BPs, and identify their severe abstract reasoning limitations.

(3) To further examine the main difficulties faced by MLLMs in solving both types of BPs (synthetic and real world ones) we introduce a focused dataset of BPs (Bongard-RWR) comprising real-world images that represent concepts from synthetic BPs using real world images. Thanks to relying on **the same abstract concepts** as synthetic BPs, Bongard-RWR facilitates direct comparisons of the MLLMs performance in both domains.

(4) We perform a detailed comparative analysis of 8 MLLMs on Bongard-RWR vs. synthetic BPs, shedding light on the reasons of their generally poor performance.

## 2  RELATED WORK

**AVR tasks.** The AVR field encompasses a broad set of problems aimed at studying various aspects of visual cognition (Gardner & Richards, 2006; Małkiński & Mańdziuk, 2023). Recent DL research in this domain gravitated towards utilizing certain well-established datasets, e.g. with visual analogies (Hill et al., 2019; Webb et al., 2020) or RPMs (Zhang et al., 2019; Barrett et al., 2018), to measure the progress of DL models. However, such benchmarks evaluate system performance in learning a particular task, rather than assessing its general ability to acquire new AVR skills. To address this limitation, certain tasks have adopted few-shot learning setups, requiring models to learn from a few demonstrations, as exemplified by SVRT (Fleuret et al., 2011) or Bongard-LOGO (Nie et al., 2020). Nonetheless, these benchmarks follow a discriminative setting where a set of possible answers is provided. Conversely, other datasets such as ARC (Chollet, 2019) or PQA (Qi et al., 2021) pose a generative challenge, which may be considered more difficult due to its open-ended nature. In addition to synthetic tasks featuring 2D geometric shapes, certain datasets present analogous reasoning tasks using real-world images (Teney et al., 2020; Ichien et al., 2021; Bitton et al., 2023). This approach extends the range of concepts that can be expressed and, above all, allows employing models pre-trained on large image datasets. In this work, we concentrate on several BP datasets that present a few-shot learning challenge, cover both synthetic and real-world images, and consider settings involving both binary classification and answer generation in natural language.

**Approaches to solve BPs.** Initial approaches to tackle BPs involved cognitive architectures (Foundalis, 2006a), program synthesis coupled with inductive logic programming (Saito &

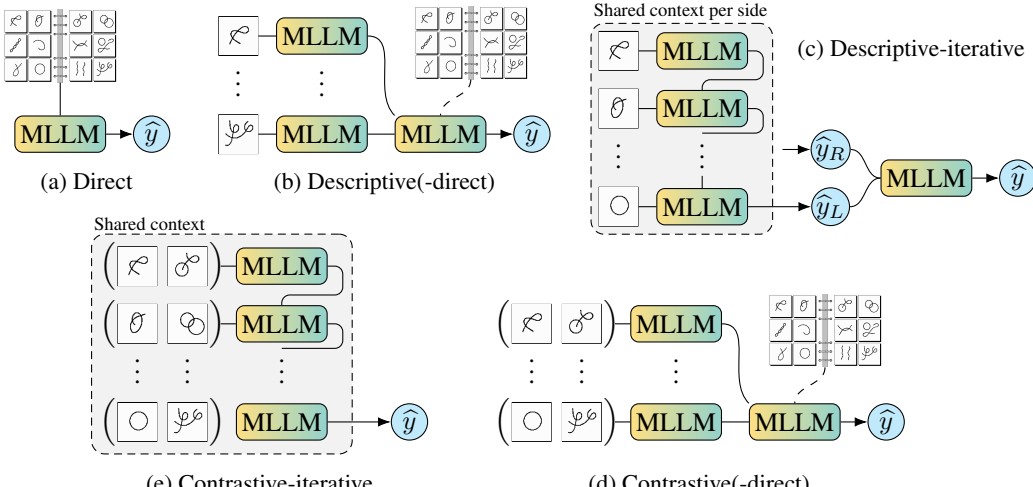

Figure 2: **Generation strategies.** Direct (a) feeds the image of the whole matrix to the model. Descriptive (b), Contrastive (d), and their iterative variants, (c) and (e), present individual image panels to the model in a fixed order. Their direct variants, (b) and (d), additionally include the image of the whole matrix. Grey background marks a sequence of requests run in a single context window.

Nakano, 1996; Sonwane et al., 2021), and the application of Bayesian inference within a visual language framework (Depeweg et al., 2018; 2024). Kharagorgiev (2018) trained a convolutional network on a generated synthetic dataset with geometric shapes and applied a one-level decision tree to solve BPs framed as a binary classification task. Nie et al. (2020) introduced Bongard-LOGO with synthetically generated BPs and used it to evaluate CNN-based models focused on meta-learning (Snell et al., 2017; Mishra et al., 2018; Lee et al., 2019; Raghu et al., 2020; Chen et al., 2021) and relational reasoning (Barrett et al., 2018). Jiang et al. (2022) applied the Relation Network (Santoro et al., 2017) to objects detected with Faster R-CNN (Ren et al., 2015) and employed the model to solve matrices from the real world Bongard HOI dataset. Despite high diversity of approaches, none of them has fully addressed the abstract and open-ended nature of BPs. Most related to our work, Wu et al. (2024) considered hybrid approaches that caption each image panel with an image-to-text model and applied LLMs for processing these text descriptions. Differently, in this work we focus on MLLMs that are inherently capable of jointly processing images and text.

**Abstract reasoning of MLLMs.** MLLMs haven't been yet applied to tackle BPs, though they were applied to several related tasks. Initial works focused on LLMs and evaluating their abstract reasoning performance in simplified analogy tasks. Webb et al. (2023) showed that GPT-3 and GPT-4 (text-only variants) performed on the human level, or even outcompeted humans, in certain RPM-like tasks in a zero-shot manner without additional fine-tuning. However, they represented the image objects as text using a fixed small vocabulary, thus omitting the need for identifying concepts from open-ended shapes, a key challenge of BPs. Recent research concerning the evaluation of abstract reasoning skills of LLMs concentrates around the Abstraction and Reasoning Corpus (ARC) task (Chollet, 2019). It was demonstrated that LLMs can solve certain ARC problems transformed to the text domain (Moskvichev et al., 2023; Mirchandani et al., 2023; Camposampiero et al., 2023; Xu et al., 2024). Despite these important stepping stones, the text-based representation taken in these works simplifies the perception task by presenting the model with pre-existing higher level representations. Only recently, thanks to the appearance of MLLMs, vision and text started to be treated jointly in a unified manner. Cao et al. (2024) proposed a suite of AVR tasks to compare MLLM and human performance. Jiang et al. (2024) assessed AVR skills of MLLMs on an introduced multidimensional benchmark combining AVR and perceptual questions. Our work complements this stream of research by exploring BPs, a fundamental task in the field, and providing insights into MLLM analogy-making performance in synthetic and real-world domains.

# 3 SOLVING BPs WITH MLLMs

In this paper, we propose a set of novel strategies for solving BPs using MLLMs. Definition of each strategy includes the input on which the model operates and the sequence of reasoning steps performed by the model. A high-level overview of these methods is provided in Fig. 2. In the main tested setting, we follow the initial BP formulation that requires providing **an answer in natural language**, and propose a model-based approach to automatically evaluate such model predictions. In addition, we consider simpler formulations of the problem, casting it into a binary classification framework that enables detailed evaluation of AVR abilities of the tested MLLMs. An illustration of these evaluation settings is presented in Fig. 3. In what follows, let $\mathcal{BP}^X = \{\mathcal{L}^X, \mathcal{R}^X, y^X\}$ denote a BP instance ($X \in \mathcal{N}$ is an index), composed of $\mathcal{L}^X = \{L_1^X, \ldots, L_6^X\}$ left and $\mathcal{R}^X = \{R_1^X, \ldots, R_6^X\}$ right panels, resp., and its concept $y^X$ expressed in natural language.

## 3.1 PROMPTING STRATEGIES FOR NATURAL LANGUAGE ANSWER GENERATION

We start by defining the strategies for generating answers in natural language. In each strategy, the model receives a general description of Bongard Problem with two BP examples with correct answers. Additionally, besides this generic introductory information, a given task $\mathcal{BP}^X$ to be solved is presented in a **strategy-specific** way. Appendix I.4 presents the exact prompt formulations.

**Direct** (Fig. 2a). The model receives an image presenting $\mathcal{BP}^X$ and is asked to directly formulate an answer (i.e., describe the difference between $\mathcal{L}^X$ and $\mathcal{R}^X$ panels in natural language).

**Descriptive** (Fig. 2b). Defines a more granular approach in which the model is first requested to generate a textual description of each image panel of the matrix. Each description is generated in a separate context, such that the model doesn't have access to the prior panels nor to their descriptions. Next, the model is requested to provide an answer to the problem based only on the generated textual descriptions of all image panels.

**Descriptive-iterative** (Fig. 2c). Evaluates the role of the reasoning context and utilizes a context window comprising the dialog history concerning all images in the given side of the problem. After generating the description of the first image, the model iteratively refines its output based on subsequent images from the same side. Based on the textual descriptions of both sides of the problem, the model is requested to provide the final answer.

**Descriptive-direct** (Fig. 2b with a dashed element). In both above Descriptive strategies, the model is never presented with the image of the whole matrix $\mathcal{BP}^X$. Descriptive-direct strategy extends Descriptive by providing the image of $\mathcal{BP}^X$ along with the textual panel descriptions.

**Contrastive** (Fig. 2d). A critical aspect of BPs is the focus on forming concepts within the specific context of the matrix $\mathcal{BP}^X$. It's often the case that correct identification of the concept governing one side requires analysis of the other side to identify their key differences. In Descriptive strategies, the model provides image descriptions concerning a single problem side $\mathcal{L}^X$ or $\mathcal{R}^X$ without taking into account the images from the other side. Conversely, in the Contrastive strategy, the model is tasked with describing the difference between a pair of corresponding images from both sides of the problem $(L_1^X, R_1^X), \ldots, (L_6^X, R_6^X)$. After describing the differences between all six image pairs in separate contexts, the model generates its final answer based on these textual descriptions.

**Contrastive-iterative** (Fig. 2e). Extends Contrastive by performing all reasoning steps in a single context window, enabling the model to gradually improve its understanding of the rule separating both sides.

**Contrastive-direct** (Fig. 2d with a dashed element). Extends Contrastive by including the image of the whole matrix together with textual descriptions of differences within each panel pair.

## 3.2 EVALUATION OF SOLUTIONS EXPRESSED IN NATURAL LANGUAGE

The correct answer to a BP may be formulated in natural language in many different ways. To account for this inherent variability, we utilize a model-based approach to assess whether the generated answer $\widehat{y}$ matches the ground-truth $y$. In the proposed setting, an MLLM ensemble receives both $\widehat{y}$ and $y$ and is requested to output a binary yes/no answer whether both descriptions refer to the

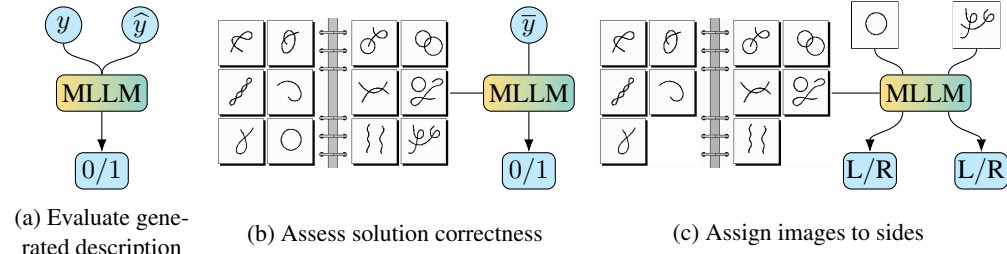

(a) Evaluate gene­rated description  (b) Assess solution correctness  (c) Assign images to sides

Figure 3: **Evaluation settings.** We consider the three settings to solve BPs: (a) the ground-truth answer $y$ is paired with a description $\widehat{y}$ generated by the MLLM and the model needs to verify if they describe the same concepts; (b) given a possible solution $\overline{y}$ the model needs to assess whether it's correct; (c) two test images corresponding to the left (L) and right (R) sides, resp., are randomly shuffled, and the model needs to assign each image from the pair to the proper side of the problem.

same concept (see Fig. 3a). Specifically, we assess the correctness of $\widehat{y}$ using all four considered proprietary MLLMs and count it as correct if at least two models agree with this class (see details in Appendix E). In contrast to solving the BP task, which requires abstract reasoning abilities, the task of determining whether two answers refer to the same concept boils down to assessing the semantic similarity between two texts, and MLLMs are known to excel in such setup (Lu et al., 2024).

### 3.3 BINARY CLASSIFICATION FORMULATIONS

To dive deeper into the AVR capabilities of the studied models, we cast the BP task into three binary classification settings, reducing the task's difficulty. Firstly, we provide the model with the image of the whole matrix $\mathcal{BP}^X$ along with a possible solution, and the model is prompted to generate a binary score assessing the correctness of the provided answer (Fig. 3b). Two settings are considered, in which the solution is formed by either the actual ground-truth answer (the expected answer is *yes*), or by an incorrect answer taken from a different BP matrix (the expected answer is *no*). Secondly, we follow a setting exemplified in the Bongard-LOGO dataset (Nie et al., 2020) in which two test images have to be classified to different sides of the problem (Fig. 3c). To this end, we take two test images corresponding to the respective sides of the matrix, randomly shuffle the images, and request the model to determine the side to which each image belongs. In synthetic BPs we create the test set by removing the 6th image from each side of the matrix, while in BPs from Bongard HOI, Bongard-OpenWorld and Bongard-RWR we use the additional test images. We refer to these three formulations as Ground-truth, Incorrect Label, and Images to Sides (see prompts in Appendix I.3).

### 4 BONGARD-RWR: SYNTHETIC BPS EXPRESSED IN REAL-WORLD IMAGES

One of the interesting research avenues is to compare the MLLMs performance on synthetic BPs vs. real-world ones. Note, however, that a direct performance comparison on synthetic Bongard dataset vs. real-world Bongard HOI and Bongard-OpenWorld datasets is not meaningful, as these datasets depict **different concepts**. To enable a meaningful comparison and additionally determine whether the MLLMs performance score is domain-related, we introduce Bongard Real-World Representa­tions, a focused dataset that expresses concepts present in synthetic BPs using real-world images, thus creating their real-life equivalents, as illustrated in Fig. 1b. Appendix F contains additional examples. The dataset is available at: `https://github.com/iclr6466/bongard-rwr`.

### 4.1 BONGARD-RWR DATASET GENERATION

For a given instance $\mathcal{BP}^X$, we first use GPT-4o to describe its underlying concept $y^X$ in $N = 10$ different ways using the prompt listed in Prompt 1. We obtain $N$ real-world textual descriptions $D_i^X = \{D_i^{XL}, D_i^{XR}\}$, $i = 0, \ldots, N - 1$, of each side $S \in \{L, R\}$. Then, we use image search engine Pexels API (Pexels, 2024) to download $M = 15$ images per each described side $D_i^{XS}$. We employ GPT-4o (see Prompt 5 in Appendix I.1) to select only those images that properly illustrate the concept of the respective side and are indeed distinguishable from the alternative concept. We

stop the selection procedure after having a set of $T = 3$ descriptions $\{D_{i_1}^X, D_{i_2}^X, D_{i_3}^X\}$, each with 2 appropriate images: $I_k^{XS}(1), I_k^{XS}(2)$, $k = i_1, i_2, i_3$ per each side $S$ (6 left and 6 right ones).

The corresponding real-world problem instance $\mathcal{RWR}^X = \{\mathcal{L}^X, \mathcal{R}^X\}$ is constructed as follows (see Algorithm 1 in Appendix F): $\mathcal{S}^X = \{I_{i_1}^{XS}(1), I_{i_2}^{XS}(1), I_{i_3}^{XS}(1), I_{i_1}^{XS}(2), I_{i_2}^{XS}(2), I_{i_3}^{XS}(2)\}$, $S \in \{L, R\}$ so as to decrease the possibility of generating a problem with a trivial answer, which is highly probable if the images from a singular textual description $D_i^X$ are taken.

---

**Prompt 1**: **Initial concept-describing prompt used in construction of Bongard-RWR.**

```
Your goal is to translate a comparison concept from the geometric
↪   domain to the real-world domain. Your translations should be
↪   expressible as images.

Example:
Geometric domain: triangles vs squares
{
   "left": {
      "concept": "pyramids"
   },
   "right": {
      "concept": "rectangular buildings"
   }
}

Give <number> unique translations for the following concept as a raw
↪   JSON array of objects (same as in the example above).
<concept>
```

---

We run Algorithm 1 for the first 100 synthetic BPs. After applying the exclusion criteria, this lead to the generation of 50 instances $\mathcal{RWR}^X$. However, as we noticed through a manual inspection, some of them were not well depicting the respective problem concept. Hence, we modified the dataset in the following way: 14 problems were entirely removed and, out of the remaining 36, 24 were adjusted through a manual selection of the images that well represent the considered concept. Furthermore, we extended the dataset by adding 17 problems with manually translated concepts (i.e., with no use of GPT-4o), for which images were also selected manually, and 7 constructed by hand, i.e., by means of making photos of manually-built scenes reflecting the respective concepts.

All in all, we obtained a real-world Bongard-RWR dataset containing 60 problems, out of which 12 were generated fully automatically, 24 were generated automatically and adjusted manually afterward (manual selection of the images), 17 were composed manually (manual translation of the concept and manual selection of the images), 7 were constructed entirely manually (photos of manually-built scenes). The details are provided in Appendix F.

## 5 EXPERIMENTS

To evaluate the AVR capabilities of MLLMs, we conduct experiments in two main settings, involving 3 binary classification setups and 7 proposed generation methods. Our evaluation spans a range of MLLMs, including 4 proprietary models accessible via API: GPT-4o, GPT-4 Turbo (Achiam et al., 2023), Gemini 1.5 Pro (Reid et al., 2024), and Claude 3.5 Sonnet (Anthropic, 2024), alongside 4 open-access models run locally on an NVIDIA DGX A100 node: InternVL2-8B (Chen et al., 2024b;a), LLaVA-1.6 Mistral-7B (Liu et al., 2024b;a; Jiang et al., 2023), Phi-3.5-Vision (Abdin et al., 2024), and Pixtral 12B (MistralAI, 2024). We consider four BP datasets covering both synthetic and real-world images. Specifically, we use the first 100 manually constructed (synthetic) BPs from (Bongard, 1970), 100 problem samples from each of Bongard HOI and Bongard-OpenWorld, and all 60 instances from Bongard-RWR. Extended results are presented in Appendix C.

**Binary classification.** Fig. 4 presents the results of binary classification tasks. In the Ground-truth setting, most proprietary and some open-access models outperform a random classifier baseline. In the Incorrect Label setting, since rejecting incorrect concepts is a generally easier task, most models perform better than in the Ground-truth setup. However, the consistently shifted performance of open-access models suggests a potential bias toward agreeing or disagreeing with provided concepts. In the Images to Sides task, proprietary models demonstrate strong performance, while Pixtral stands

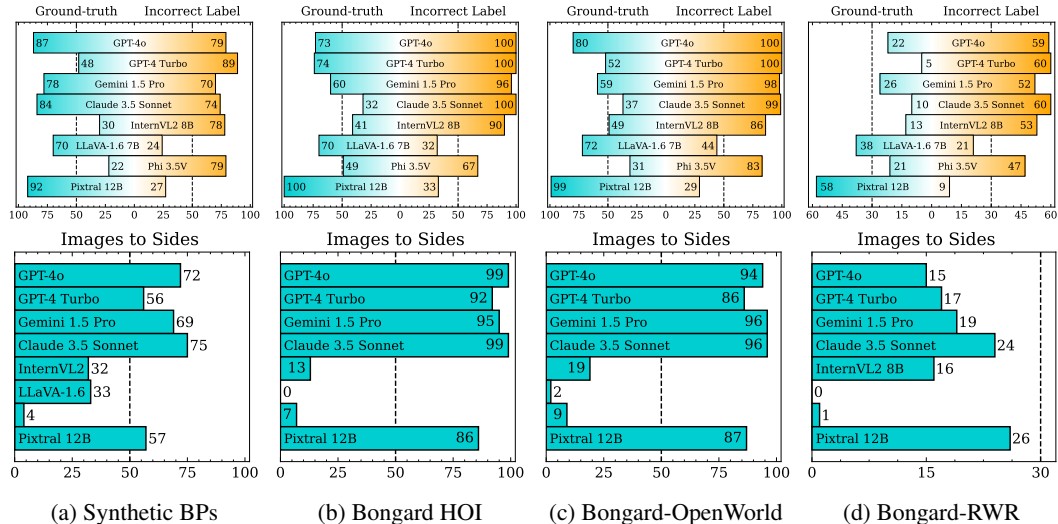

Figure 4: **Binary classification.** Results of a random baseline are marked with a dashed line.

Table 1: **Language generation.** The number of correct answers to 100 synthetic BPs, 100 selected BPs from each of Bongard HOI and Bongard-OpenWorld, and all 60 BPs from Bongard-RWR. Three main strategies: Direct, Descriptive, and Contrastive, denoted as Di, De, and Co, resp. are considered. The best result for a given strategy is marked in bold and the second best is underlined.

| | SYNTHETIC | | | HOI | | | OPENWORLD | | | RWR | | |
|---|---|---|---|---|---|---|---|---|---|---|---|---|
| | DI | DE | CO | DI | DE | CO | DI | DE | CO | DI | DE | CO |
| GPT-4O | **17** | 17 | 10 | **35** | 42 | **18** | **40** | 46 | 19 | **5** | 8 | **2** |
| GPT-4 TURBO | 6 | 15 | 8 | 22 | **45** | 5 | 21 | **57** | 12 | 1 | 5 | 0 |
| GEMINI 1.5 PRO | 7 | **21** | **17** | 23 | 40 | 15 | 13 | 32 | 11 | 3 | 7 | 1 |
| CLAUDE 3.5 SONNET | 13 | 19 | 15 | 5 | 44 | 13 | 10 | 53 | **21** | 1 | **13** | **2** |
| INTERNVL2-8B | 0 | 0 | 0 | 12 | 2 | 2 | 11 | 18 | 7 | 0 | 0 | 0 |
| LLAVA-1.6 MISTRAL-7B | 0 | 1 | 0 | 5 | 4 | 1 | 12 | 16 | 1 | 0 | 0 | 0 |
| PHI-3.5-VISION | 0 | 2 | 0 | 1 | 4 | 2 | 7 | 12 | 5 | 0 | 0 | 0 |
| PIXTRAL 12B | 1 | 4 | 1 | 28 | 27 | 7 | 33 | 34 | 14 | 1 | 1 | 0 |

out among open-access models. Nevertheless, binary classification tasks do not fully reveal whether the solver truly grasped the presented concept or simply relied on surface-level similarities, raising the need for more challenging and in-depth evaluation setups in the generative problem formulation.

**Generative capabilities in the Direct setting.** As presented in Table 1, model performance using the Direct generation strategy is generally weak on synthetic BPs, with the best model, GPT-4o, solving only 17 out of 100 problems. This indicates that the models struggle to identify abstract, synthetic concepts and express them in natural language. The challenge is even more apparent on Bongard-RWR, where the best model, GPT-4o, solves only 5 out of 60 problems. Nevertheless, performance improves on Bongard HOI and Bongard-OpenWorld, with best results of 35/100 and 40/100, resp. Notably, while GPT-4o achieves the highest scores on these two datasets, Pixtral 12B ranks second (28/100 and 33/100, resp.), showing that smaller open-access models can still be competitive in this setting. While the better performance on Bongard HOI and Bongard-OpenWorld may be attributed to a higher ratio of real-world images in the training data, the weak results on Bongard-RWR suggest that the discrepancy is more related to the **specific underlying concepts** than the visual domain as such (see Figure 13 in Appendix F for the details).

**Independent image description.** In the next experiment we analyse whether an iterative reasoning approach, in which the model first generates separate captions for each image and then combines them into a final answer, can improve performance. As shown in Table 1, compared to the Direct strategy, the Descriptive one improves the best results across all datasets: from 17 to 21 on synthetic

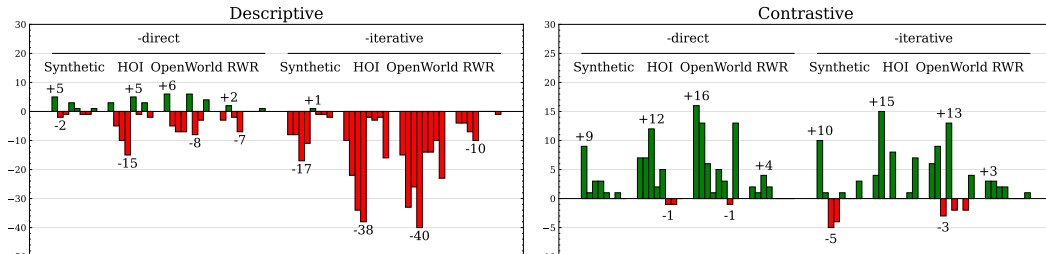

Figure 5: **The impact of -direct and -iterative variants.** Bars in each group correspond to models in the following order: GPT-4o, GPT-4 Turbo, Gemini 1.5 Pro, Claude 3.5 Sonnet, InternVL2-8B, LLaVA-1.6 Mistral-7B, Phi-3.5-Vision, and Pixtral 12B.

BPs, from 35 to 45 on Bongard HOI, from 40 to 57 on Bongard-OpenWorld, and from 5 to 13 on Bongard-RWR. A clear improvement can be observed individually for each proprietary model. The gain is less pronounced (if at all) for individual open-access models. We hypothesize that this improvement is due to the Descriptive setting being more aligned with the model's training, where it primarily learns to caption individual images. However, this strategy doesn't leverage the additional information present in the joint BP image, and certain context-dependent visual features may be missed in captioning. We believe that with further advancements in reasoning over multi-part compositional images, models in the Direct setting should eventually outperform the Descriptive strategy.

**Contrastive reasoning.** Correct identification of concepts in BPs requires a joint processing of the images from both sides of the problem. The Contrastive strategy evaluates the model ability to extract underlying differentiating concepts within such image pairs. Across all datasets, the models evaluated under the Contrastive strategy perform worse than with the Descriptive strategy (cf. Table 1). This points to the fundamental difference between human and machine approaches to solving AVR tasks. Humans often rely on direct comparisons between image panels from different categories to highlight differences (Nüssli et al., 2009), whereas the tested methods perform better when making comparisons on text-based image descriptions, potentially disregarding critical visual details missed during image captioning. This discrepancy indicates the need for further modeling improvements to fully leverage the Contrastive strategy.

**Iterative reasoning.** Next, we tested whether preserving responses from past turns in the dialog context could improve concept identification in both Descriptive and Contrastive settings. As shown in Fig. 5, the Descriptive-iterative strategy visibly worsens the results compared to its non-iterative counterpart across all datasets and models, except for negligible improvement of InternVL2-8B on synthetic BPs and several cases of a complete failure (accuracy of 0) for both strategies. In contrast, Contrastive-iterative brings no improvement over Contrastive in only 5 cases, 2 for synthetic BPs, and 3 regarding Bongard-OpenWorld. Despite these improvements, Contrastive-iterative generally performs worse than Descriptive (see Table 2, Appendix C). This indicates that contemporary models have difficulties to effectively use additional information from the context window.

**Multimodal answer generation.** In the final experiment, we assessed whether incorporating an image of the entire matrix at the answer generation step would improve the performance of the Descriptive and Contrastive strategies. As shown in Fig. 5, Descriptive-direct shows performance improvements over Descriptive in 12 out of 32 (dataset, model) cases. Contrastive-direct improves upon Contrastive in all (dataset, proprietary model) configurations, and additionally improves in certain (dataset, open-access model) settings. However, despite these gains, Contrastive-direct overall performs worse than Descriptive, except for GPT-4o and InternVL2-8B on synthetic BPs, and InternVL2-8B on Bongard HOI (see Table 2, Appendix C). This suggests that contemporary models are to some extent capable of utilizing additional visual inputs to improve reasoning performance, in particular the newest GPT-4o, which displays improvement from incorporating the -direct extension in 7 out of 8 cases. Nevertheless, further work is needed to improve consistency across all models.

**Comparison of prompting strategies.** Across all models, the Descriptive strategy achieves the highest scores on Bongard-RWR and Bongard-OpenWorld. In Bongard HOI, it ties with Descriptive-direct, while in synthetic BPs, it ranks just behind its -direct extension. As shown in

Appendix G, altogether Descriptive strategies solve the same number of synthetic BPs as Contrastive strategies (44; Fig. 15), but lead in Bongard HOI (82 vs. 63; Fig. 17), Bongard-OpenWorld (90 vs. 76; Fig. 19), and Bongard-RWR (20 vs. 11; Fig. 21). This overall advantage of Descriptive over Contrastive strategies indicates that current MLLMs perform better with prompting strategies focused on processing single images. This also highlights the need to improve multi-image reasoning capabilities of MLLMs for tasks that require reasoning across multiple images. Figs. 15 – 22 in Appendix G further show that altogether the considered approaches solved 54, 89, 93, and 23 problems from synthetic BPs, Bongard HOI, Bongard-OpenWorld, and Bongard-RWR, resp. This raises the question of whether an ensemble combining all proposed strategies could further enhance model reasoning performance. We leave the exploration of this emerging direction for future work.

**Proprietary vs. open-access models.** Proprietary models generally outperform open-access ones, leading in 35 out of 40 (dataset, strategy) pairs (see Table 2). The black-box nature of proprietary models makes it challenging to attribute their advantage to specific aspects, whether it be the number of parameters, the size and composition of training data, or the pre- and post-processing methods. However, the recently released Pixtral 12B model performs competitively in multiple settings, occasionally surpassing proprietary models. This highlights the viability of developing competitive MLLMs without sacrificing accessibility. At the same time, a clear performance drop of Pixtral 12B on synthetic BPs and Bongard-RWR suggests its intrinsic weakness in reasoning about *abstract* concepts, whether reflected in synthetic or real-world manner, similarly to other open models.

**Comparison with state-of-the-art.** A direct comparison with the results from (Wu et al., 2024) is challenging due to the different ranges of test problems used in each study. With this caveat, we concentrate on key high-level observations from both works. Wu et al. (2024) primarily focus on a binary classification setting corresponding to the Images to Sides setup in our work. On Bongard-OpenWorld, our best performing models, Gemini 1.5 Pro and Claude 3.5 Sonnet, achieved 96% accuracy, while their top method, SNAIL—a meta-learning approach leveraging pre-trained Open-CLIP image representations—achieved 64%. This suggests that MLLMs, which uniformly process images and text, outperform decoupled two-stage approaches, which handle image captioning and text-based reasoning with different models. They also briefly consider a natural language generation task, where models describe concepts presented in the BP instance (Wu et al., 2024, Appendix E). They again use a two-step approach comprising fine-tuned BLIP-2 for image captioning and ChatGPT for concept generation. In contrast, we employ a single MLLM for both tasks. For evaluating free-form concept generation, they use automated text-based metrics, which provide a general measure of text similarity. We, however, employ a voting MLLM ensemble, offering a more direct assessment of solution correctness.

**Human results.** We conducted a study with 30 participants, as detailed in Appendix B. Humans solved from 23 to 59 problems, with average of 39.2, achieving 65% accuracy. Notably, the lowest number of problems solved by a human participant (23) exceeded the number of problems solved by all models in total (22, see Fig. 22), highlighting the need for further advances in this area.

## 6 CONCLUSIONS

This paper investigates the reasoning capabilities of proprietary and open-access MLLMs using BPs as a case study. Despite rapid progress, MLLMs still exhibit significant reasoning limitations. Across all proposed answer generation strategies, the best-performing model solved only 22 out of 100 synthetic BPs. On the other hand, model performance improved moderately with real-world concepts, as shown by the results on Bongard HOI and Bongard-OpenWorld. To delve deeper into the performance discrepancies between synthetic and real-world domains, we introduced Bongard-RWR, a new BP dataset designed to represent concepts from synthetic BPs via real-world images. Focused experiments with this dataset suggest that the models' weak performance on synthetic BPs is not domain-specific but rather indicative of broader limitations in their reasoning abilities. Specifically, MLLMs struggle with recognizing abstract concepts, fail to benefit from a human-like multi-image reasoning approach, demonstrate limitations in utilizing context window effectively, and require further work to consistently integrate text and vision modalities at the answer generation step. On a positive note, experiments conducted in three binary classification settings show that some models achieve encouraging results, suggesting that current limitations may be overcome with future advancements.

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

# A    LIMITATIONS AND FUTURE WORK

**Going beyond Bongard Problems.**    BPs fundamentally require solvers to articulate answers in natural language, making them a valuable testbed for assessing the reasoning capabilities of MLLMs. However, to comprehensively explore the challenges posed by the AVR domain, it is crucial to consider a broader range of problems.

For instance, VCog-Bench (Cao et al., 2024) is a benchmark designed to evaluate the reasoning capabilities of MLLMs across 3 datasets: 560 problem instances from RAVEN (Zhang et al., 2019), 309 from CVR (Zerroug et al., 2022), and 480 from MaRs-VQA. These datasets present multi-class classification tasks, offering between 4 to 8 options per problem. While the classification setting in VCog-Bench differs from our focus on natural language generation, both studies echo a shared conclusion – MLLMs struggle in complex, multi-image reasoning tasks. We argue that generative problem formulations, such as those used in our study, pose a more substantial challenge than discriminative tasks, in which the solution may be induced from correlations or by making educated guesses. Further advances in abstract reasoning may require the development of new AVR benchmarks with generative evaluation settings.

A compelling example of a generative problem formulation is the Abstraction and Reasoning Corpus (ARC) (Chollet, 2019), in which each instance involves transforming a source grid into a target grid based on an induced transformation rule. Each instance is accompanied by a few demonstrations to guide the solver. Mitchell et al. (2023) explored multi-modal reasoning of GPT-4V on ConceptARC (Odouard & Mitchell, 2022), a variant of ARC categorizing tasks into distinct types. The study employed 3 prompting strategies: presenting all demonstrations in a single image, using separate images for each source and target grid pair, and separating each grid pair into distinct images. These settings are related to the Direct, Contrastive, and Descriptive strategies from our study, resp. The model performed best with the last approach, which aligns with the leading performance of the Descriptive strategy in our paper. Their study revealed that ARC tasks pose a significant challenge for MLLMs, aligning with our results. Similar to our findings, GPT-4V evaluation on ConceptARC demonstrated that generative problem formulations pose a significant challenge for contemporary MLLMs.

**Fine-grained analysis of MLLM perception.**    Related studies emphasize the importance of evaluating fine-grained aspects of model performance in visual reasoning tasks. Notably, Biscione et al. (2024) propose the MindSet: Vision toolbox, which categorizes tasks into three main domains: low-and mid-level vision, visual illusions, and shape and object recognition. This benchmark is specifically designed to test models on 30 psychological findings inspired by human visual perception, providing a framework for understanding similarities and differences in human and machine vision. Preliminary evaluations using ResNet-152 and GPT-4 on selected tasks revealed notable differences in perception between humans and machines. Applying MLLMs and the reasoning strategies proposed in our work to the MindSet: Vision toolbox opens a promising direction for future research, which could offer deeper insights into the perceptual capabilities of MLLMs.

**Incorporating proposed strategies to enhance abstract reasoning abilities.**    Galatzer-Levy et al. (2024) compared the cognitive abilities of MLLMs to humans using the Wechsler Adult Intelligence Scale (WAIS-IV) (Wechsler, 2008). Their findings reveal that while MLLMs excel in tasks related to verbal comprehension and working memory, they significantly underperform in perceptual reasoning tasks. The evaluation setting used in this study involved presenting models with an image of an abstract reasoning matrix alongside a text prompt describing the task, closely aligning with the Direct strategy employed in our work. However, as discussed in Section 5, the Direct strategy poses notable challenges for MLLMs. Our experiments show that models consistently achieve better performance with alternative approaches, such as the Descriptive strategy. This highlights the importance of selecting appropriate strategies when evaluating MLLMs on abstract reasoning tasks. We believe that the diverse suite of strategies proposed in our work can be extended to other studies in abstract reasoning to fully capitalize on MLLM reasoning capabilities.

**Cross-domain analysis of MLLM perception.**    A possible hypothesis for the subpar performance of MLLMs on AVR tasks involving synthetic datasets, such as VCog-Bench or ARC, is the limited representation of synthetic images in their training data. This assumption is supported by the ob-

served performance gap between synthetic BPs and real-world image BPs, such as those in Bongard HOI and Bongard-OpenWorld, which might suggest that MLLMs perform better at abstract reasoning with real-world images. However, our experiments with Bongard-RWR challenge this notion. Despite using real-world images, Bongard-RWR demonstrates that MLLMs still struggle with abstract reasoning, indicating that the performance gap cannot be solely attributed to differences in data domains. Instead, this suggests more fundamental challenges in visual reasoning. Future work could extend this research line by leveraging datasets that include both synthetic and real-world images, such as Raven's Progressive Matrices (Zhang et al., 2019; Teney et al., 2020) or Visual Analogy Problems (Hill et al., 2019; Bitton et al., 2023). Contrasting MLLM performance on such dataset pairs may provide valuable insights into whether their limitations are rooted in data domain or in broader domain-free reasoning challenges.

## B  HUMAN PERFORMANCE ON BONGARD-RWR

**Foundation of the study.**  Our tests on MLLMs using the Bongard-RWR dataset revealed their poor performance in solving synthetic concepts depicted in real-world images. However, the difficulty and reliability of this new dataset remains an open question. To address this issue, we decided to assess human capabilities in solving these problems.

**Methodology.**  We compiled all Bongard-RWR problems into a single document, including a brief introduction that explains what BPs are (see Prompt 2), along with a few detailed examples. The examples included one problem from the original BPs (#133), one from Bongard-OpenWorld, and an additional BP (#336) manually translated to the real-world domain. Bongard-RWR problem instances were positioned randomly in the document and were posed in an open-ended manner, allowing participants to provide any response they deemed appropriate.

Participants in our human evaluation predominantly belonged to the academic community, including Master students and (ocassionally) faculty members and PhD students, primarily due to accessibility. This demographic was selected based on the ease of reaching and engaging with individuals who are readily available in academic settings.

All answers were collected using an online form, ensuring a streamlined and efficient process for submission. Each participant was allowed to make only a single submission, to maintain the integrity and reliability of the data. In addition, the form contained a few more questions to gather basic statistics on our new dataset and the quality of submissions:

1. How would you assess the readability of the images included in the problems? (Scale 1-10)

2. How would you assess the difficulty of the tasks you received? (Scale 1-10)

3. What is your level of education? (Primary, Secondary, Higher, I prefer not to say)

4. How much time did you spend solving the tasks? (Less than 30 minutes, From 30 minutes to an hour, From one to two hours, More than two hours)

**Answers evaluation.**  In contrast to the evaluation of MLLM solutions, human responses were evaluated entirely manually. Initially, two humans reviewed the complete set of answers independently, achieving a 94.5% agreement on the correctness of the responses. The discrepancies were then discussed and a consensus was reached leading to a single, unified evaluation.

---

**Prompt 2**: Text used as a brief introduction in human testing.

```
The presented problems represent a type of logic puzzle. Each problem
↪   consists of two sides separated by a vertical line. Each side
↪   contains six images. The task is to find a characteristic that
↪   applies to all the images on the left side but does not apply to
↪   those on the right side.

Some problems may be less obvious and require a broader perspective or
↪   focus on details. Simply comparing the general content of the
↪   images might not be enough. Answers may repeat.
```

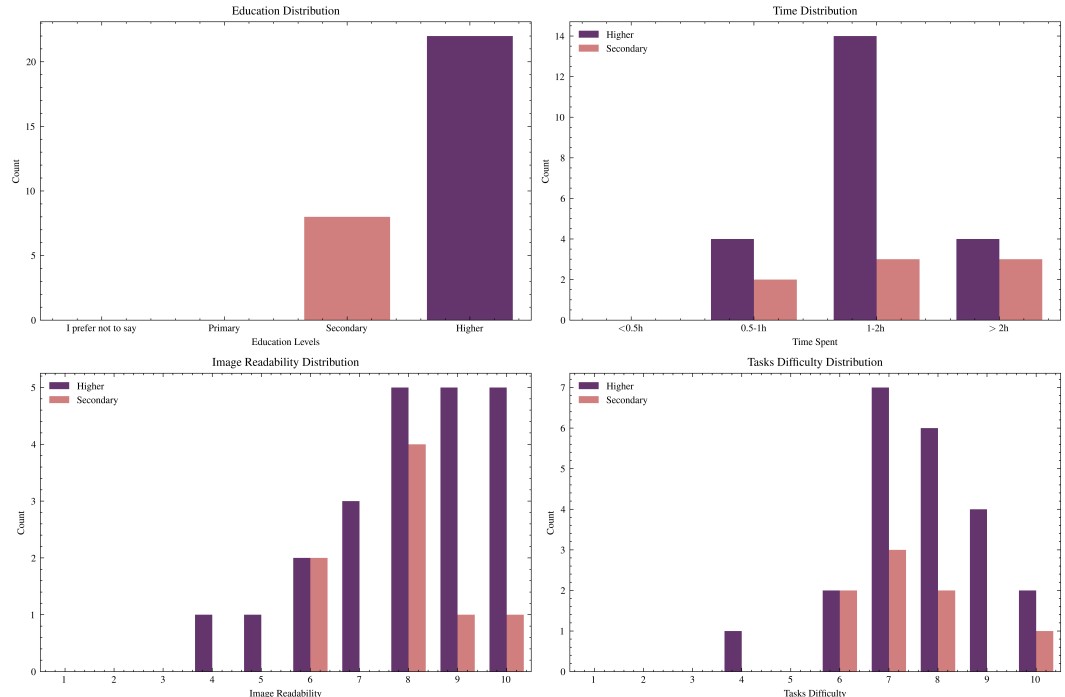

Figure 6: **The respondent's answers to the questions attached in the form.** Even though the majority of respondents had higher education, they, on average, spent more than one hour on solving the problems. Moreover, only one person rated the problems as relatively easy (giving them a difficulty score of 4 out of 10).

**Respondent Overview.** Overall, we successfully collected 30 responses, with 26.7% of participants having secondary education and 73.3% having higher education. Although the sample of respondents was small, we observed no significant discrepancies between the results of individuals with higher education and those with secondary education, both in the responses to additional questions (Figure 6 and in the problem-solving results (Figure 8).

**Findings from Respondent Responses.** In Figure 8, we present the distribution of responses to the Bongard-RWR dataset. Every problem was solved by at least one respondent, confirming the solvability of the dataset. The results are consistent across respondents (see Figure 7), with the number of solved problems ranging from 23 to 59. Moreover, the findings demonstrate the superiority of humans over MLLMs in tackling this type of task. Notably, the lowest human score exceeded the combined score of all the models. Half of the respondents solved more than 40 problems, with mean and median equal to 39.2 and 40.5, resp., resulting in 65% average accuracy.

The difficulty of each problem can be estimated based on the number of respondents who successfully solved it. As shown in Figure 8, the problems exhibit varying levels of difficulty: 22 of them were solved by at least 25 respondents, while 10 were solved by fewer than 10 respondents. In addition, three problems—10, 88 and 100—were solved by all respondents. Overall, the dataset was rated as quite difficult, with an average difficulty score of 7.6 across all respondents.

Overall, the results demonstrate the robustness and applicability of Bongard-RWR as a novel dataset for investigating the performance differences between human and model-based visual reasoning.

## C  EXTENDED RESULTS

Table 2 presents results across all models, strategies and datasets discussed in Section 5. In the following paragraphs we extend the discussion concerning binary classification settings.

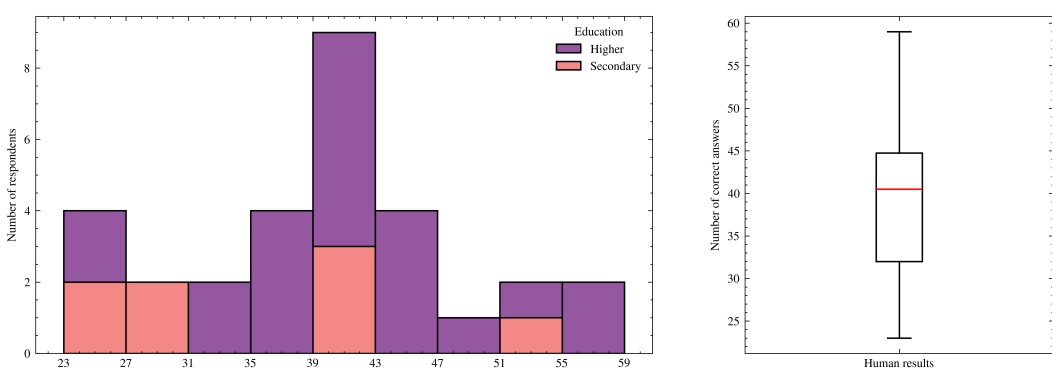

Figure 7: **The number of problems solved by humans.** The histogram illustrates the consistency across multiple respondents solving the Bongard-RWR problems. Notably, half of the respondents solved more than 40 problems, with none of them solving fewer than 23 ones. In the histogram, the lower bounds of the bins are inclusive, and the upper bounds are exclusive, except for the last bin, which is [55, 59].

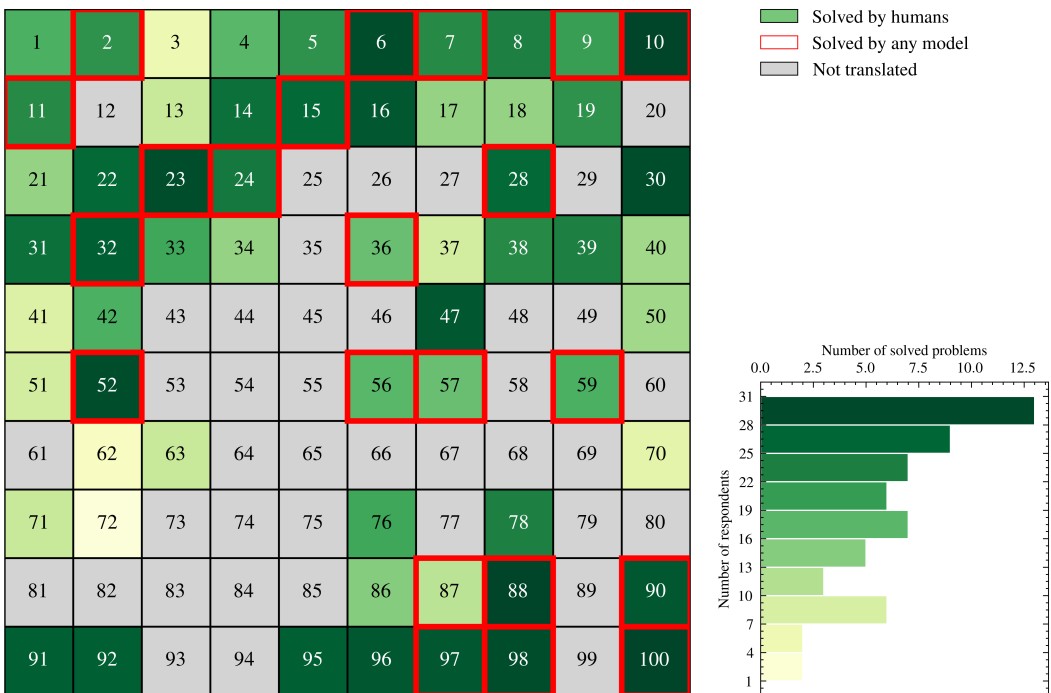

Figure 8: **Human performance on Bongard-RWR.** As shown in the plot, the models struggled with many problems that humans found relatively easy to solve. On the other hand, the models were able to solve problem #87 that appeared to be relatively demanding for human solvers. In the histogram, the lower bounds of the bins are inclusive, and the upper bounds are exclusive.

Table 2: **Evaluation results.** The number of correct answers to the first 100 synthetic BPs, 100 selected BPs from Bongard HOI and Bongard-OpenWorld, and all 60 BPs from Bongard-RWR. The best result for a given strategy is marked in bold, and the second best is underlined.

| SYNTHETIC BPS | GPT-4 O | GPT-4 TURBO | GEMINI 1.5 PRO | CLAUDE 3.5 SONNET | INTERNVL2 8B | LLAVA-1.6 MISTRAL 7B | PHI 3.5V | PIXTRAL 12B |
|---|---|---|---|---|---|---|---|---|
| GROUND-TRUTH | 87 | 48 | 78 | 84 | 30 | 70 | 22 | **92** |
| INCORRECT LABEL | 79 | **89** | 70 | 74 | 78 | 24 | 79 | 27 |
| IMAGES TO SIDES | 72 | 56 | 69 | **75** | 32 | 33 | 4 | 57 |
| DIRECT | **17** | 6 | 7 | 13 | 0 | 0 | 0 | 1 |
| DESCRIPTIVE | 17 | 15 | **21** | 19 | 0 | 1 | 2 | 4 |
| DESCRIPTIVE-ITER. | **9** | 7 | 4 | 8 | 1 | 0 | 1 | 2 |
| DESCRIPTIVE-DIRECT | **22** | 13 | 20 | **22** | 1 | 0 | 1 | 5 |
| CONTRASTIVE | 10 | 8 | **17** | 15 | 0 | 0 | 0 | 1 |
| CONTRASTIVE-ITER. | **20** | 9 | 12 | 11 | 1 | 0 | 0 | 4 |
| CONTRASTIVE-DIRECT | 19 | 9 | **20** | 18 | 1 | 0 | 1 | 1 |

| BONGARD HOI | GPT-4 O | GPT-4 TURBO | GEMINI 1.5 PRO | CLAUDE 3.5 SONNET | INTERNVL2 8B | LLAVA-1.6 MISTRAL 7B | PHI 3.5V | PIXTRAL 12B |
|---|---|---|---|---|---|---|---|---|
| GROUND-TRUTH | 73 | 74 | 60 | 32 | 41 | 70 | 49 | **100** |
| INCORRECT LABEL | **100** | **100** | 96 | **100** | 90 | 32 | 67 | 33 |
| IMAGES TO SIDES | **99** | 92 | 95 | **99** | 13 | 0 | 7 | 86 |
| DIRECT | **35** | 22 | 23 | 5 | 12 | 5 | 1 | 28 |
| DESCRIPTIVE | 42 | **45** | 40 | 44 | 2 | 4 | 4 | 27 |
| DESCRIPTIVE-ITER. | **32** | 23 | 6 | 6 | 0 | 1 | 2 | 11 |
| DESCRIPTIVE-DIRECT | **45** | 40 | 30 | 29 | 7 | 3 | 7 | 25 |
| CONTRASTIVE | **18** | 5 | 15 | 13 | 2 | 1 | 2 | 7 |
| CONTRASTIVE-ITER. | **22** | 20 | 15 | 21 | 2 | 1 | 3 | 14 |
| CONTRASTIVE-DIRECT | 25 | 12 | **27** | 15 | 7 | 0 | 1 | 7 |

| BONGARD-OPENWORLD | GPT-4 O | GPT-4 TURBO | GEMINI 1.5 PRO | CLAUDE 3.5 SONNET | INTERNVL2 8B | LLAVA-1.6 MISTRAL 7B | PHI 3.5V | PIXTRAL 12B |
|---|---|---|---|---|---|---|---|---|
| GROUND-TRUTH | 80 | 52 | 59 | 37 | 49 | 72 | 31 | **99** |
| INCORRECT LABEL | **100** | **100** | 98 | 99 | 86 | 44 | 83 | 29 |
| IMAGES TO SIDES | 94 | 86 | **96** | **96** | 19 | 2 | 9 | 87 |
| DIRECT | **40** | 21 | 13 | 10 | 11 | 12 | 7 | 33 |
| DESCRIPTIVE | 46 | **57** | 32 | 53 | 18 | 16 | 12 | 34 |
| DESCRIPTIVE-ITER. | **31** | 24 | 6 | 13 | 4 | 2 | 2 | 11 |
| DESCRIPTIVE-DIRECT | **52** | **52** | 25 | 46 | 24 | 8 | 9 | 38 |
| CONTRASTIVE | 19 | 12 | 11 | **21** | 7 | 1 | 5 | 14 |
| CONTRASTIVE-ITER. | 25 | 21 | 8 | **34** | 5 | 1 | 3 | 18 |
| CONTRASTIVE-DIRECT | **35** | 25 | 17 | 22 | 12 | 4 | 4 | 27 |

| BONGARD-RWR | GPT-4 O | GPT-4 TURBO | GEMINI 1.5 PRO | CLAUDE 3.5 SONNET | INTERNVL2 8B | LLAVA-1.6 MISTRAL 7B | PHI 3.5V | PIXTRAL 12B |
|---|---|---|---|---|---|---|---|---|
| GROUND-TRUTH | 22 | 5 | 26 | 10 | 13 | 38 | 21 | **58** |
| INCORRECT LABEL | 59 | **60** | 52 | **60** | 53 | 21 | 47 | 9 |
| IMAGES TO SIDES | 15 | 17 | 19 | 24 | 16 | 0 | 1 | **26** |
| DIRECT | **5** | 1 | 3 | 1 | 0 | 0 | 0 | 1 |
| DESCRIPTIVE | 8 | 5 | 7 | **13** | 0 | 0 | 0 | 1 |
| DESCRIPTIVE-ITER. | **4** | 1 | 0 | 3 | 0 | 0 | 0 | 0 |
| DESCRIPTIVE-DIRECT | 5 | **7** | 5 | 6 | 0 | 0 | 0 | 2 |
| CONTRASTIVE | **2** | 0 | 1 | **2** | 0 | 0 | 0 | 0 |
| CONTRASTIVE-ITER. | **5** | 3 | 3 | 4 | 0 | 0 | 0 | 1 |
| CONTRASTIVE-DIRECT | 4 | 1 | **5** | 4 | 0 | 0 | 0 | 0 |

**Binary classification (Ground-truth).** We assessed whether MLLMs can determine if a given concept matches a problem instance. On synthetic BPs, 3 proprietary (GPT-4o, Gemini 1.5 Pro, Claude 3.5 Sonnet) and 2 open-access (LLaVA-1.6 Mistral-7B, Pixtral 12B) models outperform a random classifier by a notable margin. On Bongard HOI, 3 proprietary (GPT-4o, GPT-4 Turbo, Gemini 1.5 Pro) and the same 2 open-access models also surpass random guessing. Notably, Pixtral 12B attained a perfect score on this dataset. On Bongard-OpenWorld GPT-4o, Gemini 1.5 Pro, LLaVA-1.6 Mistral-7B, and Pixtral 12B achieve reasonable results. Again, the leading model is Pixtral 12B with the outstanding 99% outcome. Model accuracy drops significantly on Bongard-RWR, where only LLaVa-1.6 Mistral-7B and Pixtral 12B outperform a random classifier. This suggests that correctly identifying concepts expressed in Bongard-RWR likely requires more advanced reasoning abilities, even in the relatively simpler binary classification setting.

**Binary classification (Incorrect Label).** Rejecting a possible solution is intuitively simpler than confirming its correctness, as it boils down to finding at least one image that doesn't match the provided concept. Accordingly, 6 models perform better in the Incorrect Label setting than in Ground-

truth, with 7 perfect scores, 2 of them on Bongard-RWR. The exceptions are LLaVA-1.6 Mistral-7B, and Pixtral 12B which are below a random guessing threshold for all four datasets, despite being above this threshold in Ground-truth. This suggests that their strong performance in the Ground-truth setting may be due to a potential bias toward agreeing with the provided concept.

**Binary classification (Images to Sides).** We also evaluate the models' ability to correctly assign two test images to the appropriate sides of the problem. A problem is considered solved if both images are correctly assigned to the respective sides. Proprietary models perform well in this task across synthetic BPs, Bongard HOI and Bongard-OpenWorld. Conversely, among open-access models, only Pixtral 12B consistently achieves strong results. Notably, on Bongard-RWR Pixtral 12B solves 26/60 problems, outperforming all proprietary models, however, all models perform below the level of random guessing. The remaining open-access models show poor results in this setting. Notably, weak results of LLaVA-1.6 Mistral-7B on real-world datasets are primarily attributed to its incorrect generation of JSON output required to format the result.

## D    THE IMPACT OF MODEL SCALING ON ABSTRACT REASONING ABILITIES

Performance of MLLMs on downstream tasks is often correlated with the number of model parameters and the size of training datasets (Kaplan et al., 2020; Hoffmann et al., 2022). To investigate the relationship between model scaling and abstract reasoning performance, we conducted experiments with a diverse set of model sizes across proprietary and open-access MLLMs. To this end, we evaluated both smaller and larger variants of the selected models. Specifically, we considered GPT-4o mini and Gemini 1.5 Flash as smaller counterparts to GPT-4o and Gemini 1.5 Pro, resp. Also, we tested multiple configurations of InternVL2 and LLaVA-NeXT model families including InternVL2-8B, InternVL2-26B, InternVL2-40B, InternVL2-Llama3-76B, LLaVA-v1.6 Vicuna-13B, LLaVA-v1.6 34B, LLaVA-NeXT 72B, and LLaVA-NeXT 110B. We conducted experiments on all 4 datasets using two solution strategies, including Direct, which is an intuitive baseline, and Descriptive, the most effective strategy identified in the main experiments.

The results are presented in Fig. 9. In general, larger proprietary models outperformed their smaller counterparts in 10 out of 16 cases. However, smaller variants sometimes performed better than larger ones. For instance, on Bongard-HOI with the Direct strategy, GPT-4o mini and Gemini 1.5 Flash surpassed their larger alternatives. This suggests that smaller models can achieve competitive performance in abstract reasoning.

For open-access models, performance consistently improved with model size. For example, the results of InternVL2 on Bongard HOI increased from 12 to 25 and from 2 to 29 for Direct and Descriptive strategies, resp. Similarly, on Bongard HOI, the performance of LLaVA-NeXT improved from 5 to 27 and from 4 to 27 for the two strategies. Analogous improvements were observed on Bongard-OpenWorld, highlighting the potential benefits of model scaling.

Despite these significant improvements in open-access models, proprietary models consistently outperformed them. In particular, GPT-4o mini achieved worse results than the best open-access model in a single case only, i.e., Bongard HOI using the Descriptive strategy (26 vs. 29). Although model scaling demonstrates its potential to enhance abstract reasoning, as shown by the open-access models, the relatively strong performance of GPT-4o mini shows that a large parameter count is not necessarily critical for excelling in abstract reasoning tasks. Consequently, these results suggest that simply scaling model size may be insufficient to achieve stronger abstract reasoning capabilities and future efforts should explicitly address this aspect, e.g., by incorporating AVR datasets into model training.

## E    EVALUATION OF MLLMS ANSWERS

Preliminary experiments revealed that proprietary MLLMs are generally much more effective in solving Bongard Problems than open, publicly-available MLLMs. Therefore, all efforts devoted to optimizing the final scores, in particular tuning the evaluation prompt were performed using these 4 commercial MLLMs.

Open-ended characteristics of BPs stemming from a textual form of an answer, and the number of considered models (8), generation strategies (7), datasets (4), and BP instances per dataset (60 in

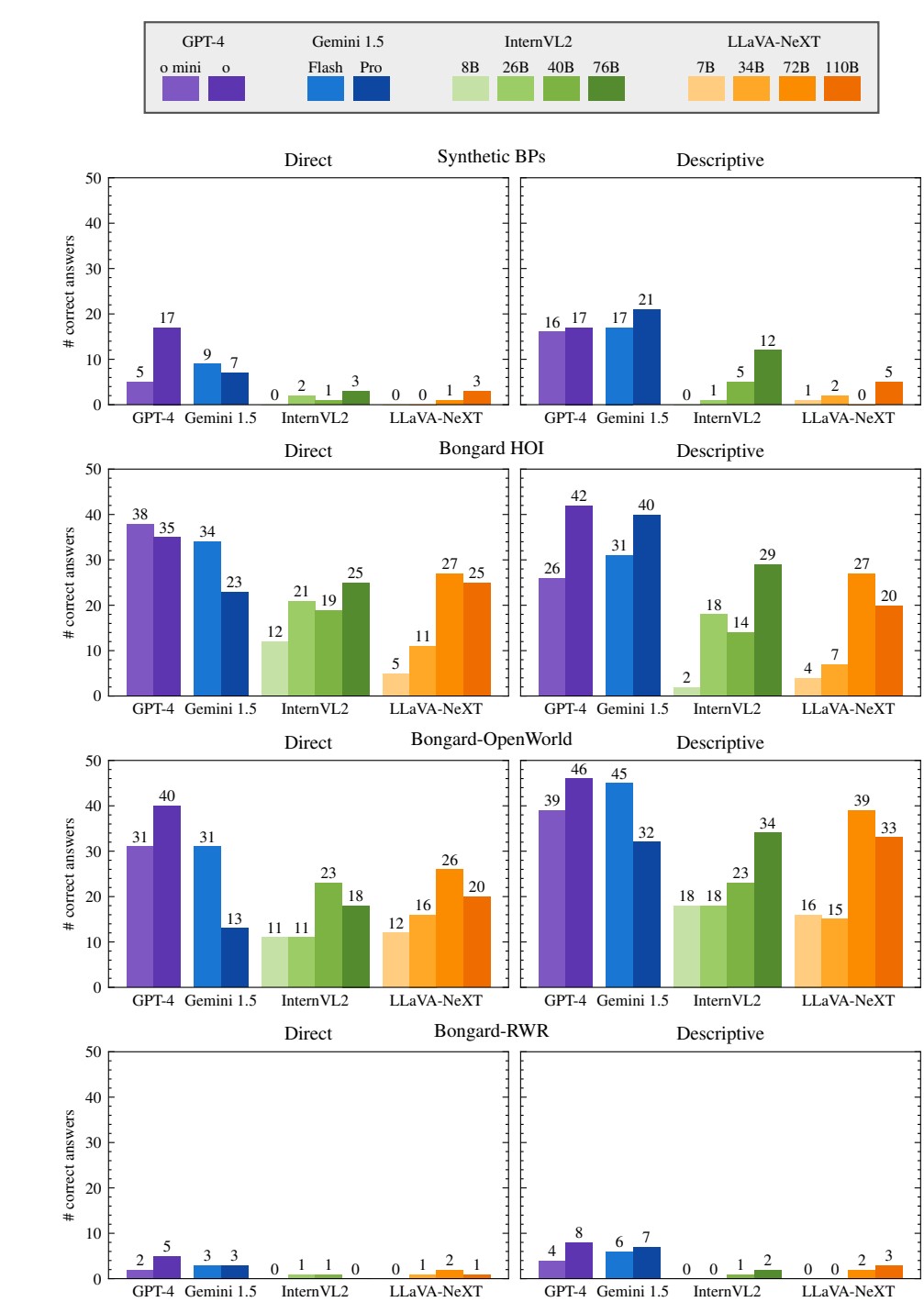

Figure 9: **The impact of model scaling on abstract reasoning abilities.** We considered a diverse set of model sizes across proprietary and open-access MLLMs. The experiments cover all 4 datasets using the Direct and Descriptive solution strategies.

Bongard-RWR and 100 in the remaining cases) require the use of an automated NLP-based evaluation of the model's answers. For this task we employed MLLMs with a specially designed prompt. The initial version of the evaluation prompt (see Prompt 3) was intentionally relatively simple – a model received an answer to be evaluated as well as the ground-truth labels, and was requested to output a binary *yes/no* answer. This prompt formulation turned out to be too simplistic. While

the level of agreement between all models was relatively high (87% of responses were rated unanimously by all models), as illustrated in Fig. 10a, manual inspection of the selected answers revealed that the assessment was generally too optimistic and relatively many evaluations wrongly pointed to *correct* answers.

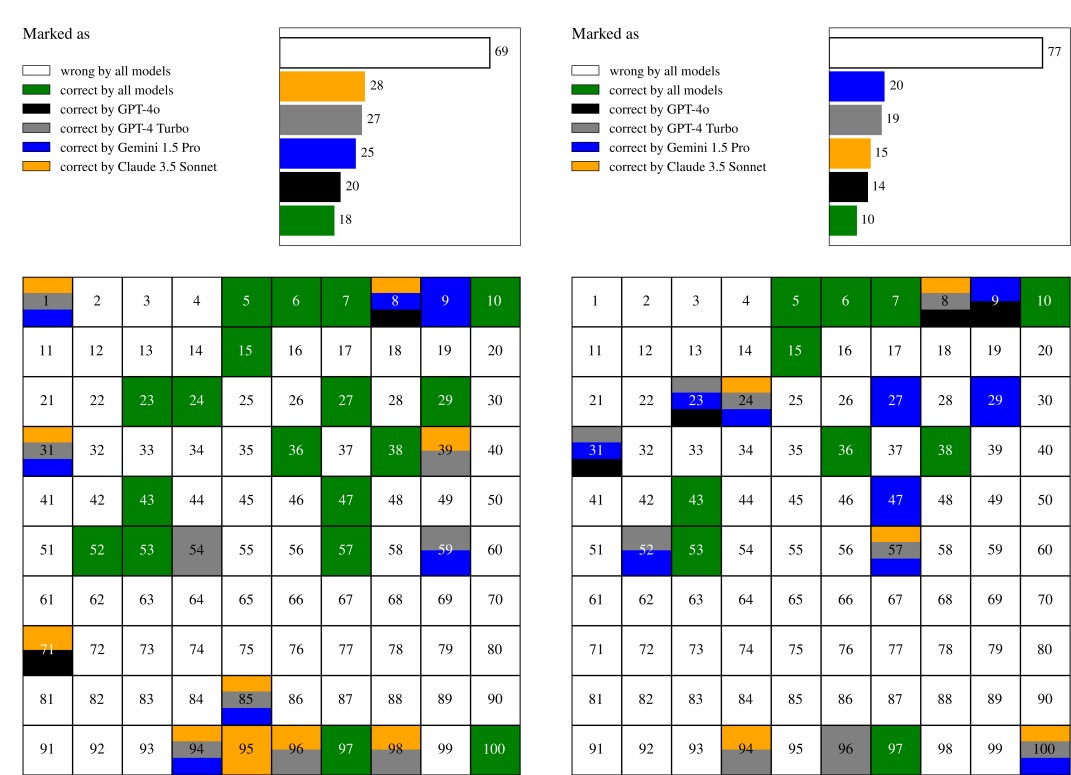

(a) Results with the **initial** evaluation prompt.   (b) Results with the **final** evaluation prompt.

Figure 10: **Models' agreement on the evaluation of BPs.** The assessed solutions were generated by GPT-4o with the *Descriptive* strategy. The numbers refer to the BP tasks from (Bongard, 1968). Green indicates tasks unanimously evaluated as correctly solved by all models, while white indicates unanimous incorrect evaluations. Other colors highlight tasks marked as correctly solved by individual models.

---

**Prompt 3**: Initial prompt used in MLLM answer evaluation. We focused on its clarity and simplicity.

```
You are a logic module designed to provide accurate answers. In a
↪   Bongard Problem the objective is to spot the difference between the
↪   contents of images located on the two opposite sides of the
↪   problem. You are given correct labels of these sides and must
↪   decide whether the answer provided by the user is correct and
↪   matches with those labels. Answer with 'OK' or 'WRONG'.

LEFT SIDE LABEL:
<left_label>

RIGHT SIDE LABEL:
<right_label>

USER ANSWER:
<model_answer>
```

## E.1 EVALUATION PROMPT OPTIMIZATION

Due to the above evaluation disagreement, we made an attempt to optimize the prompt based on the GPT-4o solutions for the additional 20 BPs (#101 – #120) that were not used in the main experiments. First, following the few-shot prompting technique, we expanded the prompt to include two examples showing a possible logical difference between correct and incorrect answers. Furthermore, we added a sentence which requested a *strict* logical compliance with the provided labels. However, this refinement appeared to be too strong, as 2 (out of 4) models didn't evaluate any of the solutions as correct.

To impose some flexibility, we changed the word *strictly* to *logically*, but this resulted in an increased rate of false-positive evaluations. Finally, we combined these two prompts, obtaining the outcome closest to the manual (our human) evaluation. The final version of our evaluation prompt is listed in Prompt 4. Additionally, we attempted to attach the image of the evaluated BP instance to each version of our prompt, but this actually confused the models rather than improving their results, so we ultimately abandoned this option and stuck with the fully text-based prompt.

Although the consistency of results regarded as the number of unanonimous assessments stayed at the same level (87%) (see Fig. 10b), the number of answers rated as correct significantly decreased, which was in accordance with our random manual verification.

Despite lowering the results variation, there were still BPs for which the assessment varied. Therefore, we eventually decided to use **hard voting** to ensemble all models' evaluations. We marked a solution as *correct* if at least 2 of the 4 models evaluated it as correct. This approach brought better results than the majority voting.

---

**Prompt 4**: **The final version of the evaluation prompt.** It is enriched with the few-shot prompting technique and imposes a logical compliance with provided labels.

```
You are a logic module designed to provide accurate answers.
In a Bongard Problem the objective is to spot the difference between
↪  the contents of images located on the two opposite sides of the
↪  problem.
You are given correct labels of these sides and must decide whether the
↪  answer provided by the user is correct and matches with those
↪  labels. Answer with 'OK' or 'WRONG'.
The user's answer has to strictly logically match the labels, as shown
↪  in examples.

FIRST EXAMPLE:
LEFT SIDE LABEL: All shapes are small.
RIGHT SIDE LABEL: All shapes are big.
USER ANSWER: On the left side, one of the shapes is small. On the right
↪  side, all of the shapes are big.
EVALUATION: WRONG
END OF FIRST EXAMPLE.

SECOND EXAMPLE:
LEFT SIDE LABEL: All shapes are small.
RIGHT SIDE LABEL: All shapes are big.
USER ANSWER: On the left side, all of the shapes are small. On the
↪  right side, all of the shapes are big.
EVALUATION: OK
END OF SECOND EXAMPLE.

LEFT SIDE LABEL:
<left_label>

RIGHT SIDE LABEL:
<right_label>

USER ANSWER:
<model_answer>
```

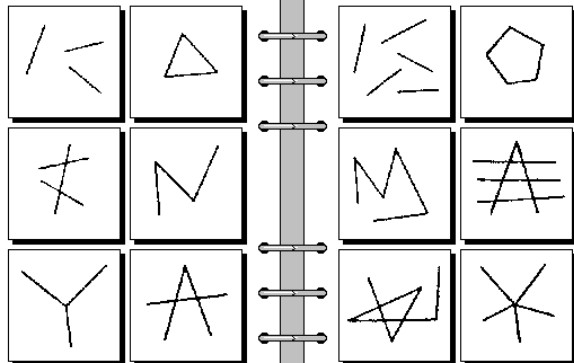

Figure 11: **Synthetic BP #85.** This is the only BP instance for which the voting-based evaluation of MLLM solutions differed from our manual evaluation. Left: Three parts. Right: Five parts. GPT-4o answer: *Left: All images are composed of exactly three lines. Right: All images are composed of more than three lines.* Voting marked it as incorrect, whereas in manual evaluation it was marked as correct. The first difference lies in the meaning of the words *lines* and *parts*, which, in this visual context, seems identical. The second difference stems from the number of the parts on the right side of the problem. The answer seems to be correct, as obviously, five is more than three. However, one could argue that the answer is incomplete, as each of the squares on the right side clearly depicts exactly five parts.

### E.2 MANUAL VERIFICATION OF THE EVALUATION PERFORMANCE OF MLLMS

In order to finally assess the efficacy of Prompt 4 we manually checked the models' evaluation performance on the 100 BPs solved by GPT-4o using the *Descriptive* strategy. As shown in Table 2 in the main paper, all proprietary models achieved better scores on incorrect labels classification. For this reason, we decided to manually verify only the problems evaluated as correct by at least one MLLM, assuming that those incorrect are generally evaluated properly. The comparison between the evaluation performance of the initial and final prompts and our manual evaluation is presented in Table 3. *All models* denotes evaluations where a solution is marked as correct only if all models evaluate it as correct. Similarly, *any model* refers to the cases where a solution is marked as correct if at least one model evaluates it as correct. *Voting* refers to the hard-voting scheme described in section E.1. It is important to observe that the chosen voting evaluation method differed from the manual evaluation in only one specific problem, which is depicted in Fig. 11.

In addition, we checked the performance of our enhanced evaluation prompt on 20 new, not used in other experiments, manually evaluated Bongard-OpenWorld problems solved by GPT-4o using the *Descriptive* strategy. Again, the use of Prompt 4 visibly increased the consensus with manual evaluation (see Table 4). The difference between our manual evaluation and the voting scheme occurred only in 2 problem solutions whose correctness is disputable (see Fig. 12).

Obviously, the choice of examples shown in the prompt may additionally impact the evaluation performance. Nevertheless, the finally proposed evaluation prompt seems to well suit both domains: synthetic and real-world, and should potentially be effective in other similar datasets and solving strategies.

| INITIAL PROMPT | | | FINAL PROMPT | | |
|:---:|:---:|:---:|:---:|:---:|:---:|
| ALL MODELS | ANY MODEL | VOTING | ALL MODELS | ANY MODEL | VOTING |
| 0.93 | 0.9 | 0.94 | 0.9 | 0.96 | 0.99 |

Table 3: **Consensus with manual evaluation on synthetic BPs.** The percentage of the solutions evaluated the same as in our manual evaluation in BP instances $\#1 - \#100$ (Bongard, 1970). The assessed solutions were obtained by GPT-4o using the *Descriptive* prompting strategy.

| | INITIAL PROMPT | | | FINAL PROMPT | |
|---|---|---|---|---|---|
| ALL MODELS | ANY MODEL | VOTING | ALL MODELS | ANY MODEL | VOTING |
| 0.75 | 0.7 | 0.7 | 0.65 | 0.85 | 0.9 |

Table 4: **Consensus with manual evaluation on Bongard-OpenWorld.** The percentage of the solutions evaluated the same as in our manual evaluation in the additional 20 Bongard-OpenWorld instances #101 − #120 (Wu et al., 2024), not used in the main experiment. The solutions were obtained by GPT-4o using the *Descriptive* prompting strategy.

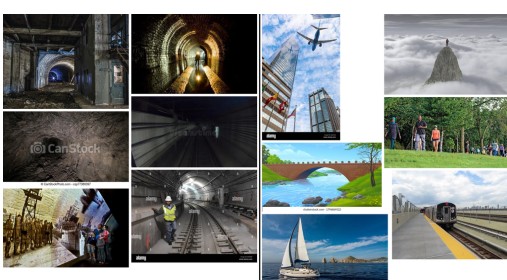

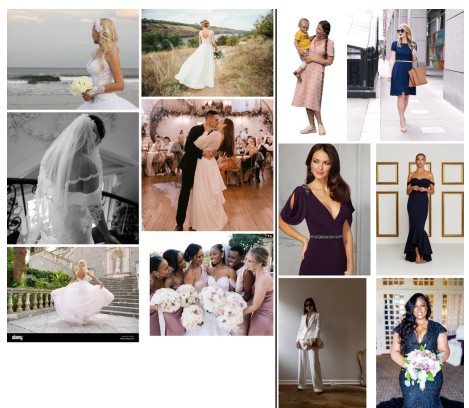

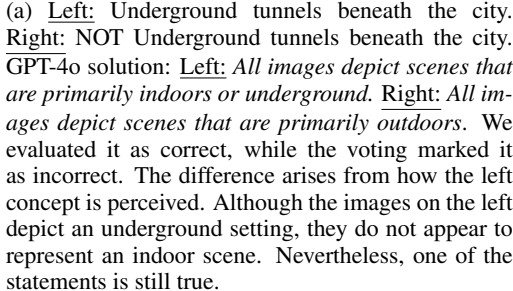

(a) Left: Underground tunnels beneath the city. Right: NOT Underground tunnels beneath the city. GPT-4o solution: Left: *All images depict scenes that are primarily indoors or underground.* Right: *All images depict scenes that are primarily outdoors.* We evaluated it as correct, while the voting marked it as incorrect. The difference arises from how the left concept is perceived. Although the images on the left depict an underground setting, they do not appear to represent an indoor scene. Nevertheless, one of the statements is still true.

(b) Left: A woman wearing a white wedding dress. Right: NOT A woman wearing a white wedding dress. GPT-4o solution: Left: *All images feature women in white wedding dresses or wedding-related scenes.* Right: *All images feature women in non-wedding attire, wearing dresses or suits of various colors other than white.* We evaluated it as incorrect, while the voting marked it as correct. The difference stems from a small detail in the right concept. One of the images on the right depicts a woman in a white suit, which conflicts with the model's answer.

Figure 12: **The only two Bongard-OpenWorld problems (out of the selected 20) for which the voting evaluation differed from our manual evaluation.** The correctness of GPT-4o's solutions to these problems is disputable.

## F    BONGARD-RWR DATASET

Bongard-RWR dataset developed in this work is attached in the **technical appendix** and will also be released for the reserach community under the MIT license. The dataset generation algorithm is presented in Algorithm 1 using notation introduced in Section 4.1.

Furthermore, Fig. 14 provides additional examples of the proposed Bongard-RWR dataset. Each subfigure presents a comparison between the synthetic Bongard problem and its respective real-world translation in Bongard-RWR. Examples 14a and 14b were translated automatically, whereas 14c and 14d were constructed fully manually, including building an appropriate scene and taking a picture. Additionally, Fig. 13 shows a particular approach taken when translating a given synthetic BP to its Bongard-RWR counterpart (problems not translated and those rejected after translation are combined into one category).

## G    COVERAGE OF BONGARD-RWR INSTANCES

Even though the final results of individual models and strategies solving Bongard-RWR are somewhat unsatisfactory, especially in the case of open language response generation, it is worth to

---

**Algorithm 1** The Bongard-RWR dataset generation.

---

**Input:** A set of synthetic concepts $Y$
**Output:** A set of generated instances $\mathcal{RWR}$

1: $M \leftarrow 15, \quad N \leftarrow 10, \quad T \leftarrow 3$
2:
3: **for** $y^X \in Y$ **do**
4: $\quad D^X \leftarrow \text{GenerateTranslations}(y^X, N)$
5: $\quad I^X \leftarrow \emptyset, \quad P \leftarrow \emptyset$
6:
7: $\quad$ **for** $D_i^X \in D^X$ **do**
8: $\quad\quad$ **for** $m \leftarrow 1$ to $M$ **do**
9: $\quad\quad\quad$ **for** $S \in \{L, R\}$ **do**
10: $\quad\quad\quad\quad I \leftarrow \text{DownloadImage}(D_i^{XS}, m)$
11: $\quad\quad\quad\quad$ **if** $I$ is accepted by model **then**
12: $\quad\quad\quad\quad\quad I_i^{XS} \leftarrow I_i^{XS} \cup \{I\}$
13: $\quad\quad\quad\quad$ **end if**
14: $\quad\quad\quad$ **end for**
15:
16: $\quad\quad\quad$ **if** $|I_i^{XL}| \geq 2$ and $|I_i^{XR}| \geq 2$ **then**
17: $\quad\quad\quad\quad P \leftarrow P \cup \{i\}$
18: $\quad\quad\quad\quad$ **break**
19: $\quad\quad\quad$ **end if**
20: $\quad\quad$ **end for**
21:
22: $\quad\quad$ **if** $|P| \geq T$ **then**
23: $\quad\quad\quad$ **Break**
24: $\quad\quad$ **end if**
25: $\quad$ **end for**
26:
27: $\quad \mathcal{L}^X \leftarrow \emptyset, \quad \mathcal{R}^X \leftarrow \emptyset$
28: $\quad$ **if** $|P| \geq T$ **then**
29: $\quad\quad$ **for** $k \leftarrow 1$ to $6$ **do**
30: $\quad\quad\quad p \leftarrow P[k \mod T]$
31: $\quad\quad\quad j \leftarrow k \div T$
32: $\quad\quad\quad$ **for** $S \in \{L, R\}$ **do**
33: $\quad\quad\quad\quad \mathcal{S}^X \leftarrow \mathcal{S}^X \cup \{I_p^{XS}(j)\}$
34: $\quad\quad\quad$ **end for**
35: $\quad\quad$ **end for**
36:
37: $\quad\quad \mathcal{RWR}^X \leftarrow \{\mathcal{L}^X, \mathcal{R}^X\}$
38: $\quad\quad \mathcal{RWR} \leftarrow \mathcal{RWR} \cup \{\mathcal{RWR}^X\}$
39: $\quad$ **end if**
40: **end for**

---

examine the degree of overlap of correct answers across the tested models. On the one hand, the results of ground-truth classification and model's disagreement on the solution evaluation clearly confirm inability of any single model to solving all problems from the Bongard-RWR datatset. On the other hand, it is likely that the overlap is not complete, and it is posible to expand the solution coverage by appropriate mixture-of-experts approach. This is indeed the case in our experiments, as illustrated in Figures 15 - 22. While single proprietary MLLMs solved between 9 and 14 instances, the number of instances solved by *any* MLLM equaled 23. We leave exploration of this path for future research.

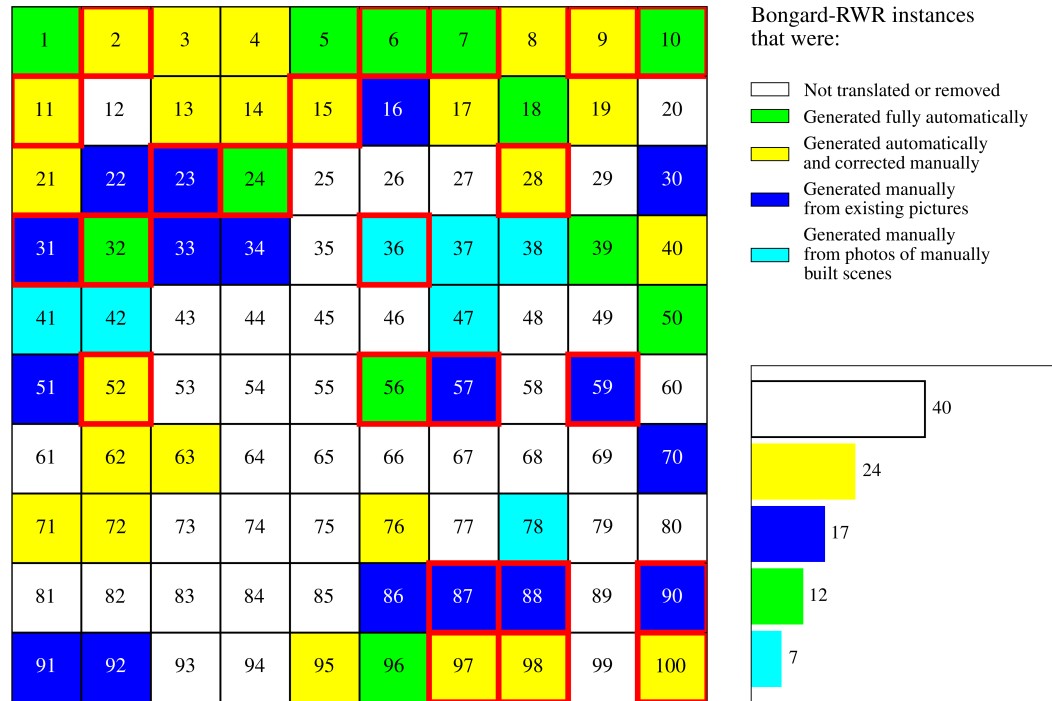

Figure 13: **The structure of the Bongard-RWR dataset.** Each color denotes the genesis of the translation of the respective problem. Problems not translated and those rejected after translation are combined into one category. Red color outlines denote problems which were solved by any model using any strategy. There is no visible correlation between the set of solved problem and the methods used for their generation.

# H   COMPARISON OF SYNTHETIC BONGARD VS. BONGARD-RWR RESULTS

Our research shows that all of the tested models have difficulty solving synthetic concepts when applied to real-world images. Comparing the results for both datasets (see Figures 16 and 22) we identified some discrepancies. Four problems that remained unsolved in the synthetic BPs were successfully solved in the real-world domain of the Bongard-RWR dataset: #56, #87, #88, and #98. However, three of these problems differ slightly from their synthetic counterparts. The images in #56 from Bongard-RWR feature a variety of colors instead of the usual black-and-white figures. Furthermore, in real-world version of problem #87 more images feature disjoint elements instead of multi-part objects, which may have nudged the model toward the correct answer. Additionally, in problem #98, the figures are shown against a hatched texture, which was not accounted for in the real-world translation.

Conversely, 22 problems that were solved in the synthetic BPs were not solved in the Bongard-RWR dataset. In most cases, the models focused on general associations that did not apply to all the images. For example, the concept presented in problem #3 is: "LEFT: Outline figures, RIGHT: Solid figures." Claude 3.5 Sonnet, using the Descriptive strategy, responded: "LEFT: focuses on practical, everyday objects or scenes, RIGHT: emphasizes aesthetic, artistic, or decorative elements that serve more for visual appeal than utility". Nevertheless, both sides feature a red coffee cup with a saucer, which matches both generated descriptions. The key difference lies in the color of the saucer's rim, which determines whether it should be considered as an outline or a solid figure.

# I MLLM PROMPTS

## I.1 PROMPTS FOR BONGARD-RWR GENERATION

Prompt 5 was used to select those images that correctly represent given concept translation. In addition to the left and right concepts, we also provided prompts briefly explaining the context that the image should match. These prompts were generated during the translation stage of our algorithm (see the fourth line in Algorithm 1).

---

**Prompt 5**: Prompt used for the selection of proper images for a translated concept.

```
You translated a concept comparison from geometric domain to the
↪  real-world domain as follows:

Geometric domain: <left_geometric_concept> vs <right_geometric_concept>
Real world domain:
{
    "left": {
        "concept": <left_concept>,
        "prompt": <left_concept_description>
    },
    "right": {
        "concept": <right_concept>,
        "prompt": <right_concept_description>
    }
}

Now, you need to check if the queried image matches your translation
↪  and provides enough information to distinguish it from the other
↪  concept. Don't focus too much on the prompt. It's just a hint for
↪  you to understand the concept better.
Provided image represents <side_concept>

Give your answer in the following format:
EVALUATION: OK
EXPLANATION: <here you can provide additional information>
or
EVALUATION: REJECTED
EXPLANATION: <here you can provide additional information>
```

---

## I.2 PROMPT DESCRIBING THE BONGARD PROBLEM

Prompt 6 describing the BP task has been placed at the beginning of each solving strategy introduced in Section 3.1.

---

**Prompt 6**: Prompt explaining Bongard problems to an MLLM.

```
A Bongard Problem is composed of left and right sides separated by a
↪  line. Each side contains six images. All images belonging to one
↪  side present a common concept, which is lacking in all images from
↪  the other side. The goal is to describe the rule that fits all
↪  images on the left side, but none on the right, and, conversely,
↪  the rule that fits all images on the right side, but none on the
↪  left. The description of the rule should be simple and concise.
Example 1: All shapes on left are small. All shapes on right are big.
Example 2: The left side contains circles. The right side contains
↪  triangles.
```

---

## I.3 PROMPTS FOR CLASSIFICATION STRATEGIES

Prompt 8 was used to assess solution correctness (see Section 3.2). Prompt 7 was used to assign images to sides (see Section 3.3).

**Prompt 7**: Prompt used for images to side classification (see Figure 3c). Two examples were provided to not bias the results of the model.

```
You are a vision understanding module designed to provide short, clear
↪  and accurate answers. Your goal is to classify two test images to
↪  the corresponding side of the Bongard Problem, LEFT or RIGHT. Each
↪  image belongs to exactly one class. The test images belong to
↪  different classes.

The images are always provided correctly. Respond only to the specific
↪  request. Respond in json using the following format.

FIRST EXAMPLE:
Left images: <small shapes>
Right images: <big shapes>

First test image: <small shape>
Second test image: <big shape>

Response:
{
    "first": {
        "explanation": "The test image shows a small shape, similarly
        ↪  as all images on the left side. Conversely, the images on
        ↪  the right side feature big shapes.",
        "concept": "small vs big",
        "answer": "LEFT"
    },
    "second": {
        "explanation": "The test image shows a big shape, similarly as
        ↪  all images on right. The images on left, on the other hand,
        ↪  feature small shapes.",
        "concept": "small vs big",
        "answer": "RIGHT"
    }
}
END OF FIRST EXAMPLE

SECOND EXAMPLE:
Left images: <circles>
Right images: <triangles>

First test image: <triangle>
Second test image: <circle>

Response:
{
    "first": {
        "explanation": "The test image shows a triangle, which matches
        ↪  all images on right. In contrast, the left side images
        ↪  feature circles.",
        "concept": "circles vs triangles",
        "answer": "RIGHT"
    },
    "second": {
        "explanation": "The test image shows a circle, which matches
        ↪  all images on left. Conversely, the right side images
        ↪  feature triangles.",
        "concept": "circles vs triangles",
        "answer": "LEFT"
    }
}
END OF SECOND EXAMPLE
```

**Prompt 8**: Prompt used for the solution correctness assessment (see Figure 3b).

```
You are a vision understanding module designed to provide short, clear
↪  and accurate answers. Your goal is to evaluate the correctness of
↪  the provided answer to the given Bongard Problem. All images are
↪  provided correctly. Do not explain the answer, just evaluate it.
↪  Respond 'OK' if the answer is correct, otherwise respond 'WRONG'.

User answer: <user_answer>
```

### I.4 PROMPTS FOR NATURAL LANGUAGE ANSWER GENERATION STRATEGIES

Prompts 9–16 were used for natural language answer generation (see Section 3.1).

**Prompt 9**: Prompt used for the *Direct* strategy. (see Figure 2a).

```
You are a vision understanding module designed to provide short, clear
↪  and accurate answers. Your goal is to solve the provided Bongard
↪  Problem. What is the difference between the two sides of the
↪  problem?
```

**Prompt 10**: Prompt used to obtain the image descriptions in the *Descriptive* strategy (see Figure 2b).

```
The provided image is a part of an abstract visual reasoning problem.
↪  Describe all crucial properties of the image. Your description
↪  should be as concise as possible. Focus on the most important
↪  details. The image is provided correctly. Respond only with
↪  descriptions.
```

**Prompt 11**: Prompt used for the *Descriptive* and *Descriptive-direct* strategies (see Figure 2b).

```
You are a vision understanding module designed to provide short, clear
↪  and accurate answers. Your goal is to solve the provided Bongard
↪  Problem using descriptions of its images.

LEFT IMAGES:
<left_descriptions>

RIGHT IMAGES:
<right_descriptions>

What is the difference between the two sides of the problem?
```

**Prompt 12**: Prompt used to obtain the image descriptions in the *Descriptive-iterative* strategy (see Figure 2c). After the last image, we used the prompt: "That was the last image. Now provide your final answer."

```
You'll receive a sequence of images that are a part of a single side of
↪  a Bongard Problem. The images will be provided one by one. Your
↪  goal is to find a common concept presented in all images. Your
↪  description should be as concise as possible. Focus on the most
↪  important details. Try to enhance the description of the concept
↪  after each image.

The image is always provided correctly. Respond only to the specific
↪  request. The first image will be provided in the next message.
```

**Prompt 13**: Prompt used for the *Descriptive-iterative* strategy (see Figure 2c).

```
You are a vision understanding module designed to provide short, clear
↪  and accurate answers. Your goal is to solve the provided Bongard
↪  Problem using descriptions of two sides of the problem.

LEFT SIDE DESCRIPTION:
<left_description>

RIGHT SIDE DESCRIPTION:
<right_description>

What is the difference between the two sides of the problem?
```

**Prompt 14**: Prompt used to obtain the comparison between the left and right image in the *Contrastive* strategy (see Figure 2d). After the last image, we used the prompt: "That was the last image. Now provide your final answer."

```
You are given two images extracted from the left and right side of a
↪  Bongard Problem, respectively. Your goal is to compare the images.
↪  Your comparison should be as concise as possible.
```

**Prompt 15**: Prompt used for the *Contrastive* and *Contrastive-direct* strategies (see Figure 2d).

```
You are a vision understanding module designed to provide short, clear
↪  and accurate answers. Your goal is to solve the provided Bongard
↪  Problem using comparisons between pairs of images. Each pair
↪  contains one image from the left and one from the right side of the
↪  problem.

COMPARISONS:
<comparisons>

What is the difference between the two sides of the problem?
```

**Prompt 16**: Prompt used for the *Contrastive-iterative* strategy (see Figure 2e). After the last pair of images, we used the prompt: "It was the last pair of images. What is the difference between the two sides of the problem?"

```
You are a vision understanding module designed to provide short, clear
↪  and accurate answers. Your goal is to solve the provided Bongard
↪  Problem. You'll receive a sequence of image pairs. Each pair
↪  contains one image from the left and one from the right side of the
↪  problem. In each step compare the two images and refine the
↪  definitions of concepts that describe left and right sides of the
↪  problem. Your description should be as concise as possible. Focus
↪  on the most important details. The first pair will be provided in
↪  the next message.
```

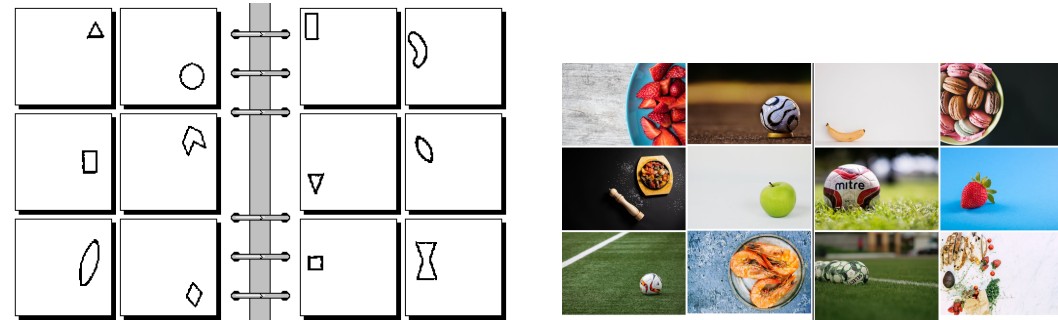

(a) Synthetic BP #8 with its automatically translated Bongard-RWR version. Left: Ends of the curve are far apart. Right: Ends of the curve are close together.

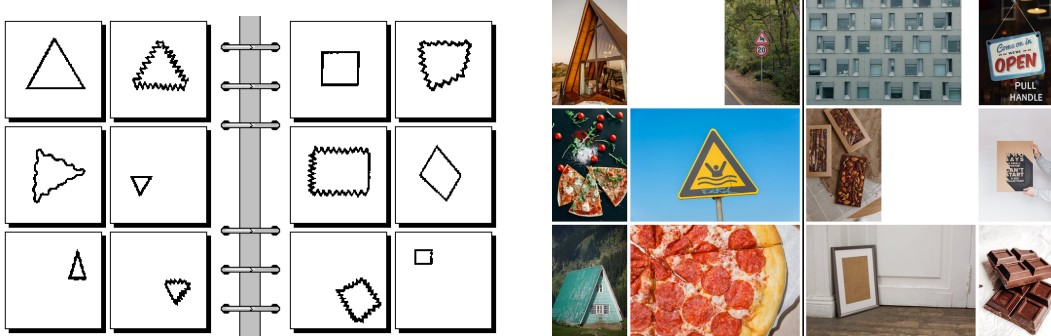

(b) Synthetic BP #10 with its automatically translated Bongard-RWR version. Left: Triangles. Right: Quadrangles.

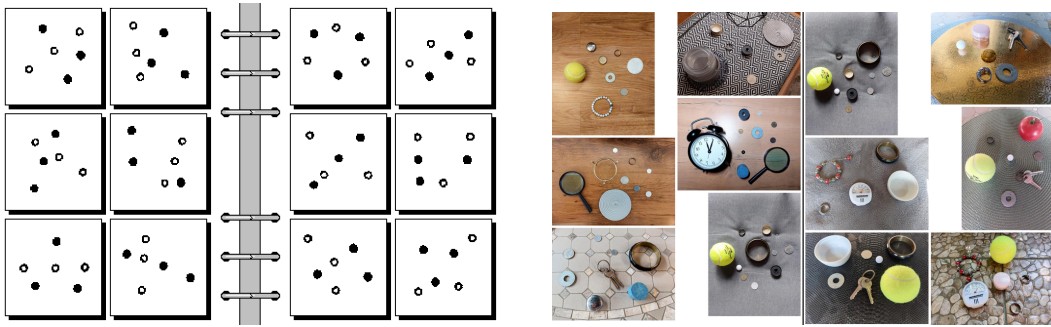

(c) Synthetic BP #41 with it manually constructed Bongard-RWR version. Left: Outline circles on one straight line. Right: Outline circles not on one straight line.

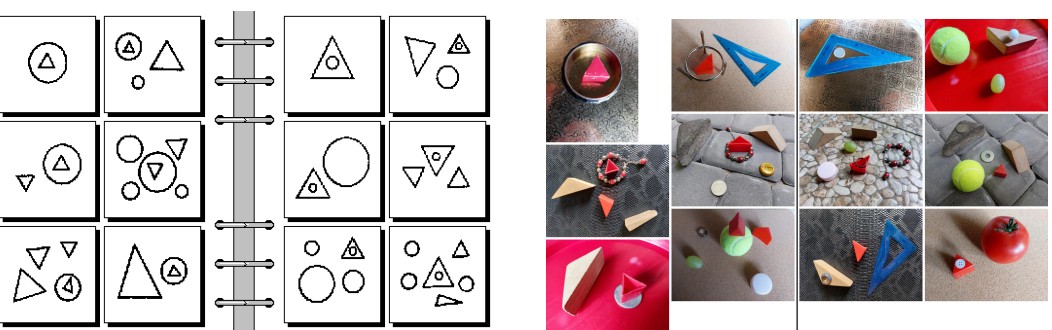

(d) Synthetic BP #47 with its manually constructed Bongard-RWR version. Left: Triangle on top of the circle. Right: Circle on top of the triangle.

Figure 14: **Additional examples of Bongard-RWR instances.**

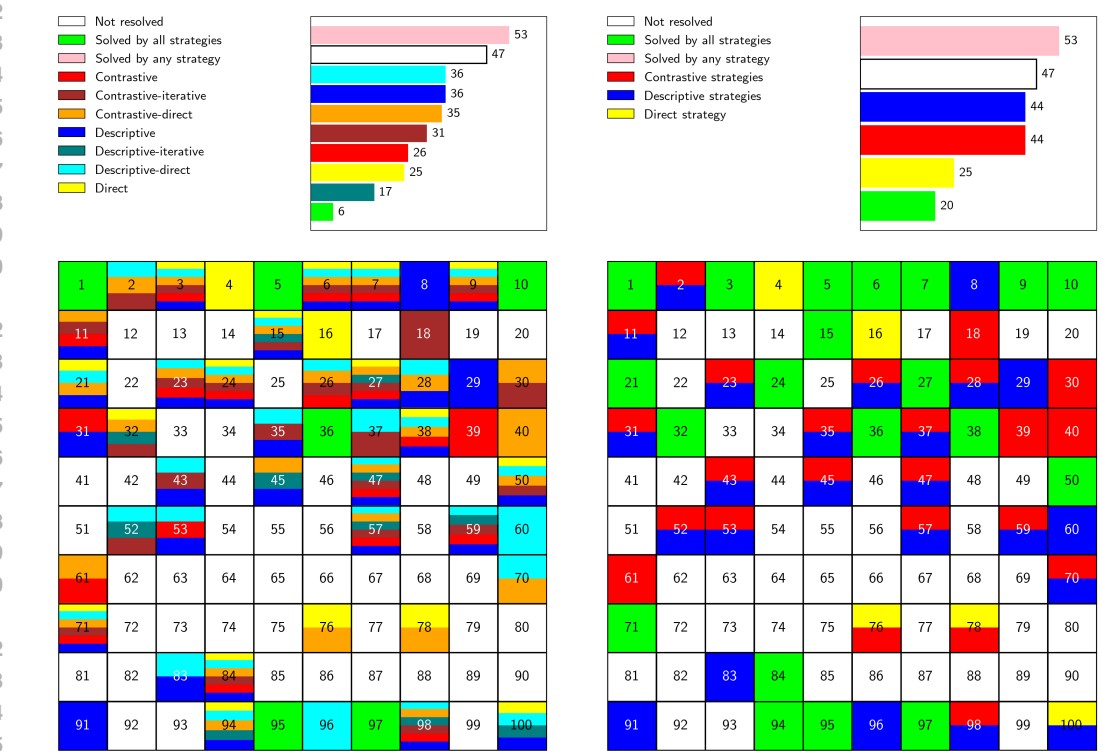

Figure 15: **Overall result of each strategy on synthethic BPs.** Colormaps depict all problems solved by any tested model using the respective prompting strategy. The right figure aggregates strategies into corresponding groups for better coverage exposure.

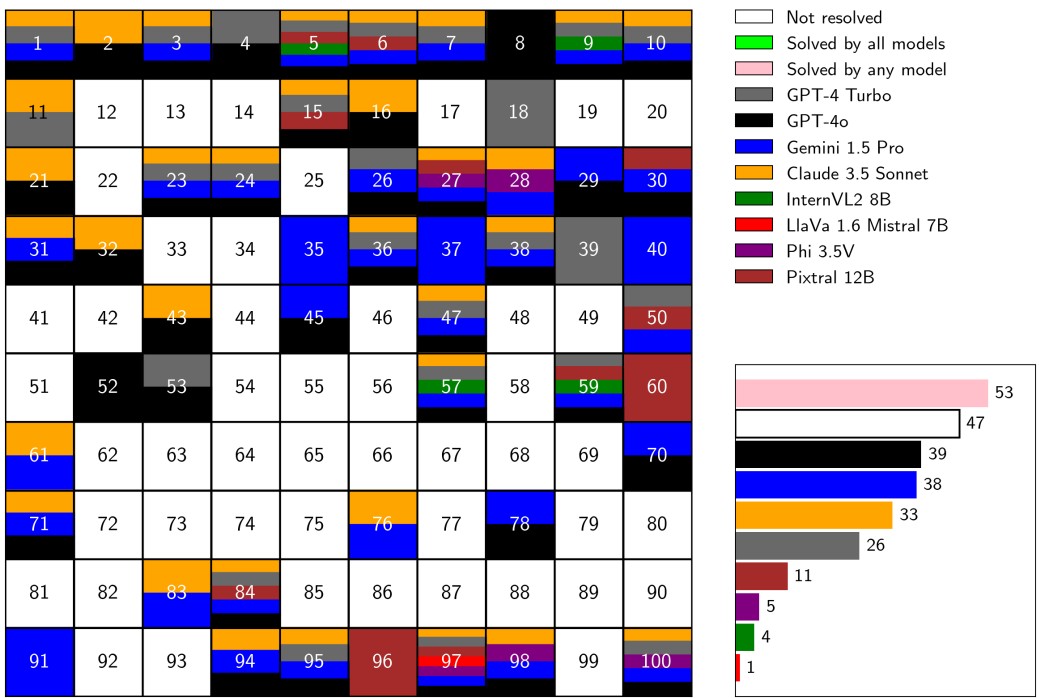

Figure 16: **Final summary of all experiments on the synthetic Bongard Problems dataset.** The colormap aggregates all problems solved by any tested model using any generation prompting strategy. Overall, the models collectively managed to solve 53 problems.

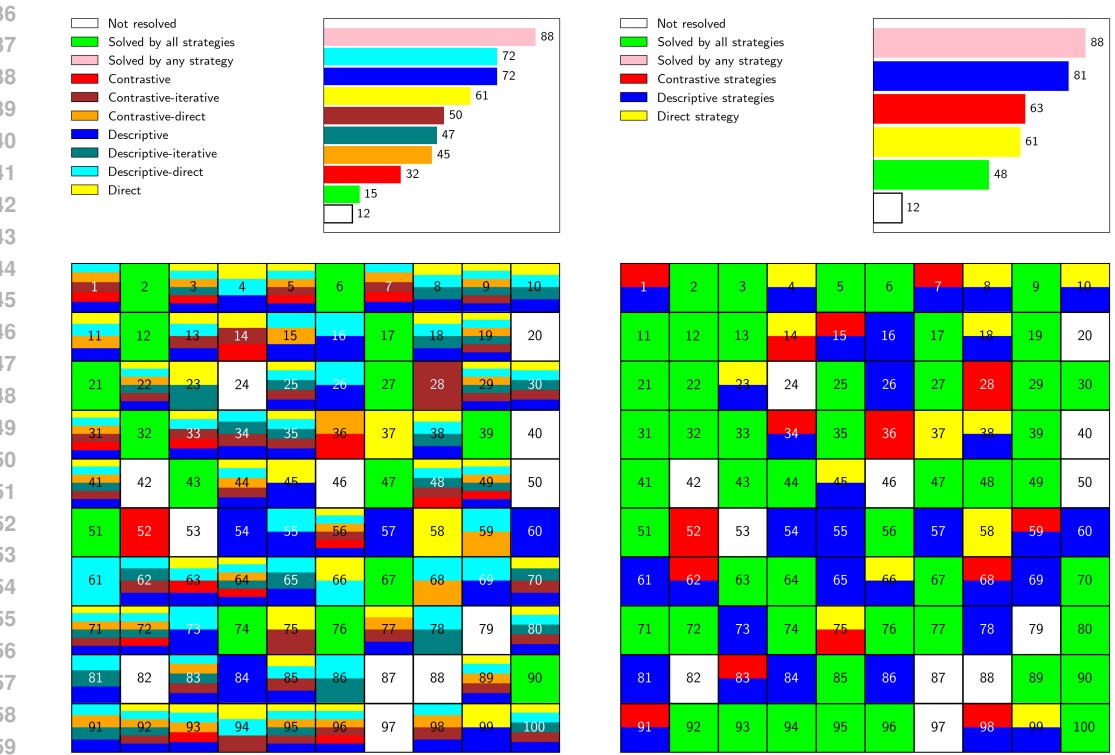

Figure 17: **Overall result of each strategy on Bongard HOI.** Colormaps depict all problems solved by any tested model using the respective prompting strategy. The right figure aggregates strategies into corresponding groups for better coverage exposure.

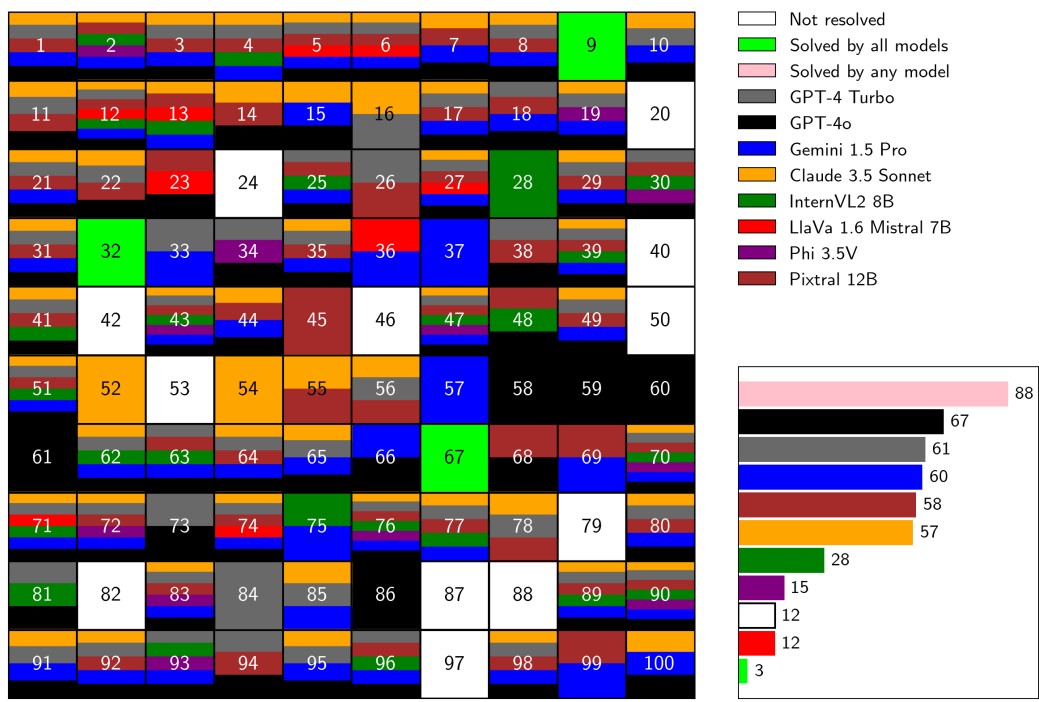

Figure 18: **Final summary of all experiments on the Bongard HOI dataset.** The colormap aggregates all problems solved by any tested model using any generation prompting strategy. Overall, the models collectively managed to solve 88 problems.

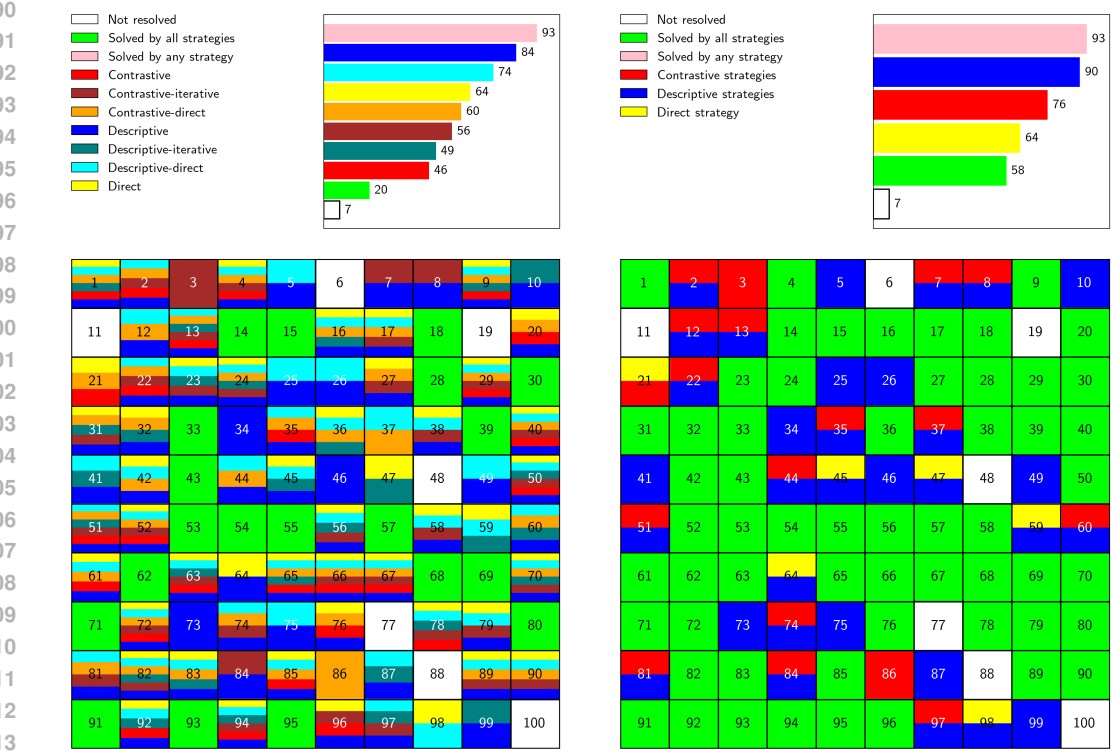

Figure 19: **Overall result of each strategy on Bongard-OpenWorld.** Colormaps depict all problems solved by any tested model using the respective prompting strategy. The right figure aggregates strategies into corresponding groups for better coverage exposure.

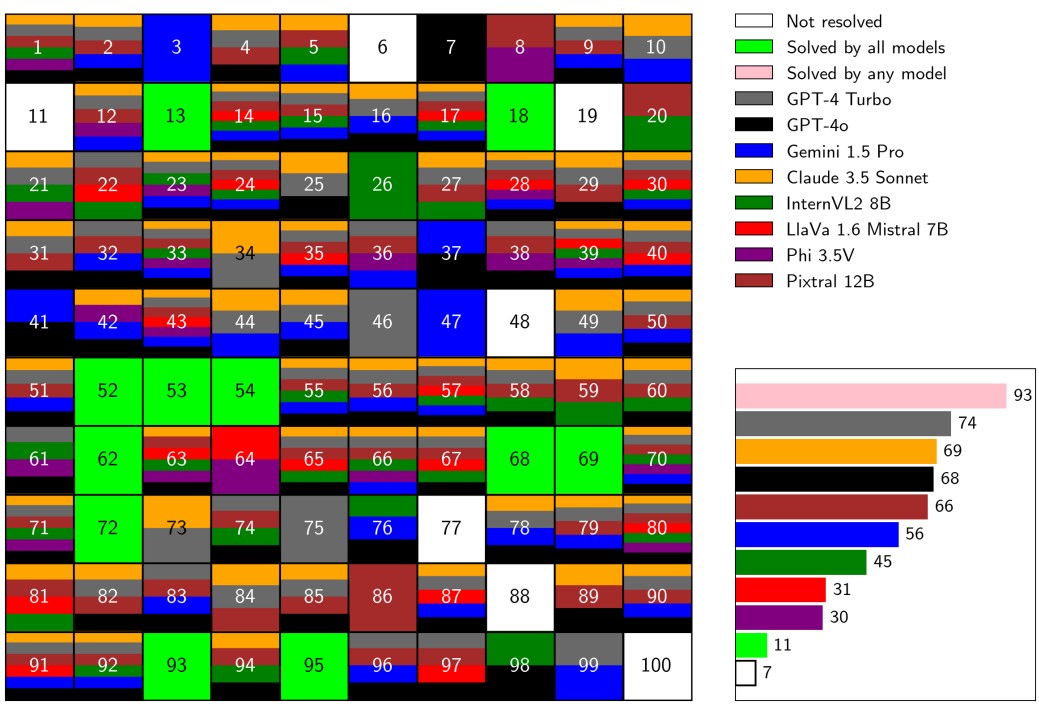

Figure 20: **Final summary of all experiments on the Bongard-OpenWorld dataset.** The colormap aggregates all problems solved by any tested model using any generation prompting strategy. Overall, the models collectively managed to solve 93 problems.

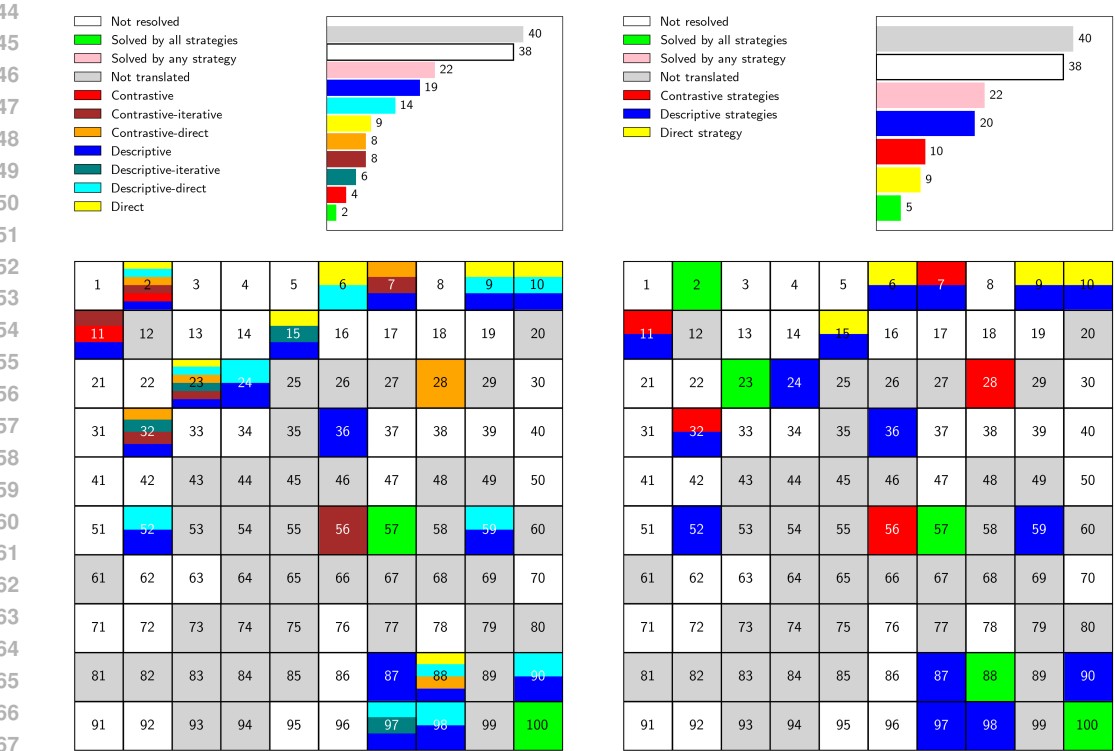

Figure 21: **Overall result of each strategy on Bongard-RWR.** Colormaps depict all problems solved by any tested model using the respective prompting strategy. The right figure aggregates strategies into corresponding groups for better coverage exposure.

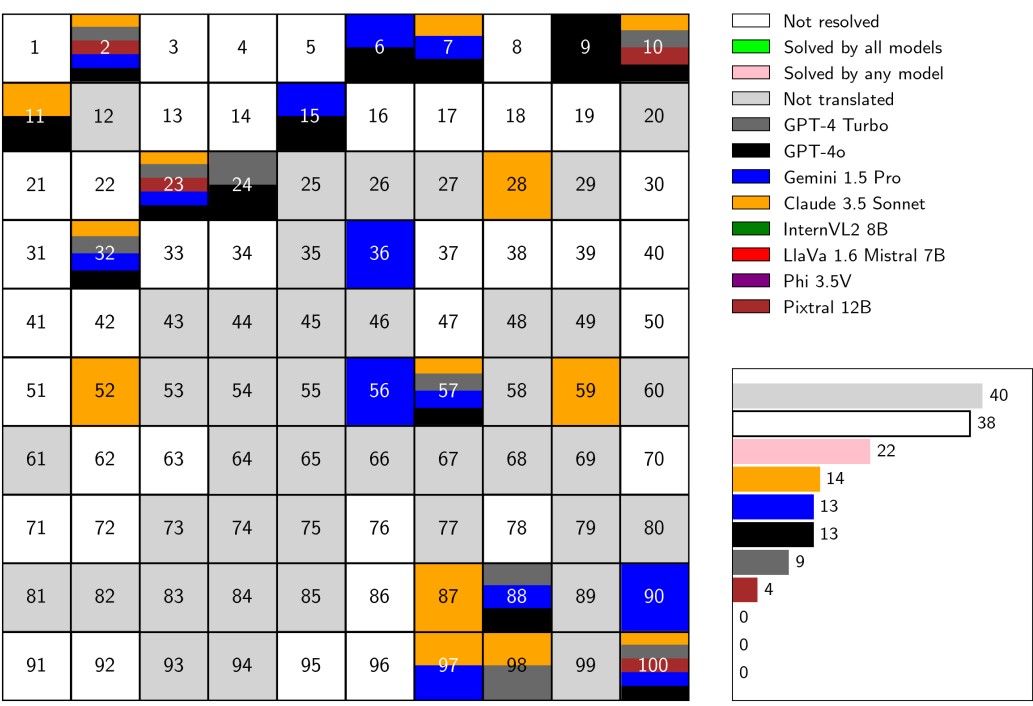

Figure 22: **Final summary of all experiments on the Bongard-RWR dataset.** The colormap aggregates all problems solved by any tested model using any generation prompting strategy. Overall, the models collectively managed to solve 22 problems.

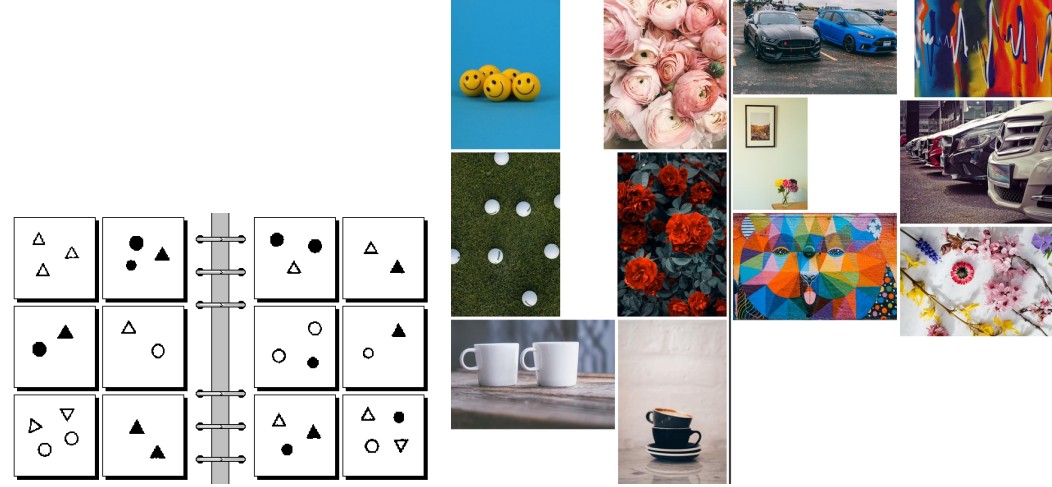

(a) Model's answer: "Left: All shapes are filled. Right: At least one shape is unfilled". Evaluated as incorrect.

(b) Model's answer: "Left: Images are monochromatic (containing only shades of a single color). Right: All images contain at least one hollow (unfilled) shape". Evaluated as correct.

Figure 23: Synthetic BP #56 with its automatically translated Bongard-RWR version. Correct answer: "Left: All figures of the same color. Right: Figures of different colors". Provided answers belong to Gemini 1.5 Pro using the Contrastive-iterative strategy, as it was the only combination that solved this Bongard-RWR instance correctly.

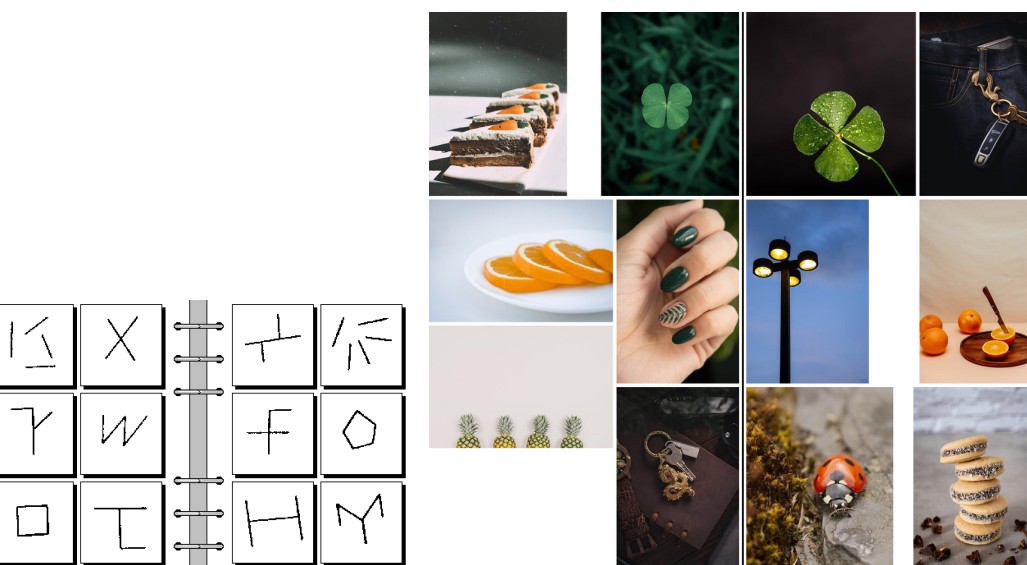

(a) Model's answer: "Left: All shapes are composed of straight lines that form only acute or right angles. Right: All shapes contain at least one instance of obtuse angles or curves (e.g., pentagon, "H" shape, radial pattern)". Evaluated as incorrect.

(b) Model's answer: "Left: All images contain exactly four of something: four orange slices, Four cake slices, four lobes of the four-leaf clover, four pineapples, four fingers with painted nails (thumb not visible), four visible elements (key, keychain, watch/compass, leather strap). Right: Images do not contain exactly four of any element: single four-leaf clover, multiple keys on a ring, single street lamp with multiple lights, single ladybug, multiple oranges (more than four), stack of five cookies". Evaluated as correct.

Figure 24: Synthetic BP #87 with its manually translated Bongard-RWR version. Correct answer: "Left: Four parts. Right: Five parts". Provided answers belong to Claude 3.5 Sonnet using the Descriptive strategy, as it was the only combination that solved this Bongard-RWR instance correctly.

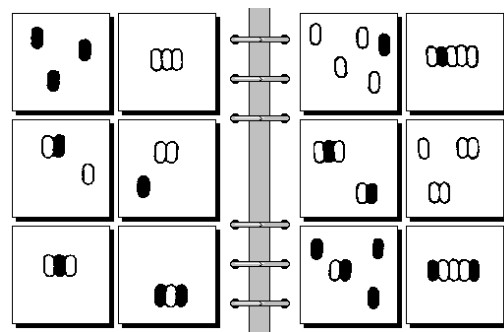 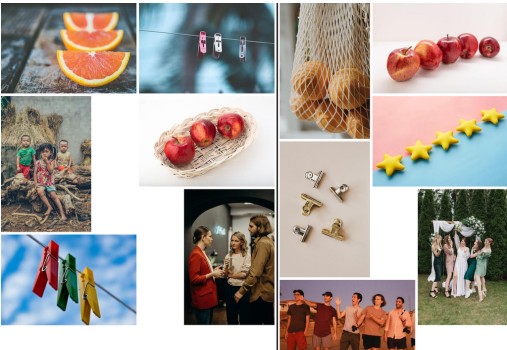

(a) Model's answer: "Left: Each image contains an even number of shapes. Right: Each image contains an odd number of shapes". Evaluated as incorrect.

(b) Model's answer: "Left: Contains groups of three items. Right: Contains groups of five items". Evaluated as correct.

Figure 25: Synthetic BP #88 with its manually translated Bongard-RWR version. Correct answer: "Left: Three parts. Right: Five parts". Provided answers belong to GPT-4o using the Direct strategy. Overall this problem was solved by three models and four strategies.

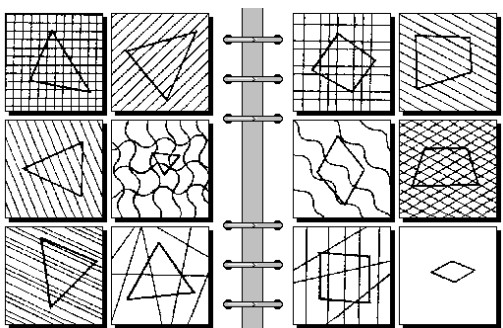 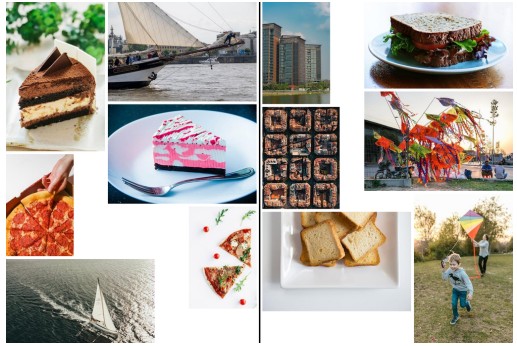

(a) Model's answer: "Left: All images contain geometric shapes or patterns that are oriented or have components that predominantly point to the right. Right: All images contain geometric shapes or patterns that are either static (not pointing in any specific direction) or have components that do not predominantly point to the right". Evaluated as incorrect.

(b) Model's answer: "Left: All images on the left side feature objects that are primarily triangular in shape. This includes the slices of cake and pizza, which are cut into triangular pieces, and the sailboats, which have triangular sails. Right: All images on the right side feature objects or scenes that are primarily rectangular or square in shape. This includes the modern buildings, the sandwich, the grid-like urban area, the arrangement of kites, and the plate of toasted bread, all of which emphasize rectangular or square forms". Evaluated as correct.

Figure 26: Synthetic BP #98 with its manually corrected Bongard-RWR version. Correct answer: "Left: Three parts. Right: Five parts". Provided answers belong to GPT-4 Turbo using the Descriptive-direct strategy. Overall this problem was solved by two models using two different strategies.

