# OpenReview forum: "Reasoning Limitations of  Multimodal Large Language Models. A case study of Bongard Problems"
_ICLR.cc/2025/Conference — Submitted to ICLR 2025_

### Official Review · Reviewer_9XwS · 2024-10-30

**Soundness:** 3
**Presentation:** 3
**Contribution:** 3
**Rating:** 8
**Confidence:** 3

**Summary:**

This paper investigates the abstract reasoning capabilities of multimodal large language models (MLLMs) using Bongard Problems (BPs), which require analogy-making between sets of images. The authors test several MLLMs on both synthetic and real-world BP datasets, highlighting significant limitations in their performance, especially with abstract, synthetic problems. Despite some improvement with real-world images, the models still struggle with effectively incorporating new information and handling complex visual reasoning tasks. To address this, the paper introduces a new dataset, Bongard-RWR, translating synthetic BP concepts into real-world images.

**Strengths:**

- The paper studies an interesting topic of Bongard Problems (BPs). It introduces a comprehensive evaluation of MLLMs using BPs, which are notoriously challenging in abstract reasoning. By testing both synthetic and real-world BP datasets, the authors highlight critical weaknesses in these models, providing valuable insight into their current capabilities. This positions BPs as a powerful benchmark for future MLLM advancements.
- This paper creates Bongard-RWR to bridge the gap between abstract synthetic reasoning and real-world tasks. This dataset allows for more meaningful comparisons of MLLM capabilities across domains. Also, this dataset enables future work to pinpoint the source of reasoning failures more precisely.
- The paper is well written and easy to understand.

**Weaknesses:**

I didn't see major weaknesses.

**Questions:**

- Some open-source models like InternVL2 and Phi-3.5V perform well on many benchmarks but struggle, nearing chance-level performance on Bongard Problems, while LLava-1.6 and Pixtral perform okay. What do the authors think causes these discrepancies?
- How closely does abstract visual reasoning (AVR) reflect real-world visual reasoning (VR)? For instance, if a model excels in AVR, can we expect it to perform well in VR, and vice versa? Some potential correlations experiments between AVR and VR performance might give some insights.

---

> ### Author Response · Authors · 2024-11-23
> **Response to the Reviewer 9XwS**
>
> > Some open-source models like InternVL2 and Phi-3.5V perform well on many benchmarks but struggle, nearing chance-level performance on Bongard Problems, while LLava-1.6 and Pixtral perform okay. What do the authors think causes these discrepancies?
>
> The open models considered in our study differ in several key aspects, including the data mixtures used during pre-training, fine-tuning and learning from human feedback, the architectures of their text and vision encoders, and the approaches for combining multimodal embeddings. These variations make it challenging to pinpoint a single factor driving the observed differences in performance. Furthermore, the nature of BPs is quite distinct from tasks typically found in standard MLLM data sources. For example, training datasets often include tasks like visual question answering  or image captioning, while analogy-based reasoning tasks like BPs receive little to no coverage, which may contribute to variability in model performance.
>
> Nevertheless, both Pixtral and LLaVA-NeXT introduce important improvements in their vision encoders. In particular, Pixtral processes images at their original resolution and aspect ratio, offering greater flexibility in the number of tokens used for image processing. LLaVA-NeXT introduces the AnyRes algorithm, which supports high-resolution image processing by segmenting images into different grid configurations. We hypothesize that high-fidelity vision encoders are critical for MLLMs to excel in solving BPs.
>
> > How closely does abstract visual reasoning (AVR) reflect real-world visual reasoning (VR)? For instance, if a model excels in AVR, can we expect it to perform well in VR, and vice versa? Some potential correlations experiments between AVR and VR performance might give some insights.
>
> Thank you for this thought-provoking question. The primary distinction between AVR and VR lies in task formulation. VR tasks are typically explicit and grounded in clear objectives, such as object detection or segmentation. For example, a common VR task in visual question answering (VQA) might involve an image of a room and a corresponding question such as “What stands next to the table on the left side?”. In contrast, AVR tasks focus on abstract analogies, which require identification of similarities and differences across images based on an underlying pattern expressed in a more conceptual and abstract manner.
>
> Despite these differences, AVR and VR share key similarities. AVR tasks fundamentally require reasoning across a set of images, which overlaps with certain VR benchmarks focused on multi-image reasoning. For instance, the MMMU [1] and MMIU [2] benchmarks contain tasks that test capabilities of reasoning across multiple images. From this perspective, datasets and strategies used in our work can serve as benchmarks to evaluate multi-image VR capabilities of MLLMs.
>
> Our experiments reveal an important relationship between AVR and VR performance. Proprietary MLLMs pre-trained on diverse datasets and fine-tuned on various VR tasks demonstrate relatively strong performance in our AVR experiments. However, good VR performance does not necessarily guarantee strong AVR capabilities, as evidenced by the results of open-access models. For instance, InternVL2-8B reported a relatively strong performance of 51.8% on MMMU [3], which is a good result even when compared to a much larger model such as Claude 3.5 Sonnet, which achieved 68.3% [4]. In contrast, InternVL2-8B performed significantly worse than Claude 3.5 Sonnet in our experiments (e.g. on Bongard HOI using the Descriptive strategy the models scored 2% and 44%, resp.). This discrepancy indicates that strong VR performance does not necessarily transfer to AVR problems.
>
> Vice versa, excelling in AVR does not necessarily imply strong performance in VR. For instance, a supervised classifier trained specifically to solve synthetic BPs is unlikely to generalize to real-world scenarios. Assessing model’s applicability to VR tasks requires employing AVR datasets that span both synthetic and real-world domains, such as those considered in our work. A model that performs well across such diverse setup is more likely to demonstrate generalizable reasoning abilities applicable to the broader VR domain.
>
> [1] Yue, Xiang, et al. "MMMU: A Massive Multi-discipline Multimodal Understanding and Reasoning Benchmark for Expert AGI." Proceedings of the IEEE/CVF Conference on Computer Vision and Pattern Recognition. 2024.
>
> [2] Meng, Fanqing, et al. "MMIU: Multimodal Multi-image Understanding for Evaluating Large Vision-Language Models." arXiv preprint arXiv:2408.02718 (2024).
>
> [3] OpenGVLab. “InternVL2-8B Model card.” https://huggingface.co/OpenGVLab/InternVL2-8B Accessed: 2024-11-21.
>
> [4] Anthropic. “Claude 3.5 Sonnet.” https://www.anthropic.com/news/claude-3-5-sonnet Accessed: 2024-11-21.

---

### Official Review · Reviewer_VpcT · 2024-11-05

**Soundness:** 2
**Presentation:** 3
**Contribution:** 2
**Rating:** 3
**Confidence:** 4

**Summary:**

This paper investigates the Bongard Problems (BPs) as a case study in Multimodal LLMs. This problem represents a foundational problem in Abstract visual reasoning (AVR). This paper gathered a new dataset Bongard-RWR and used multiple strategies to test the proprietary and open-sourced MLLMs on both synthetic and real-world BPs. The experiments show that there is still a gap for MLLMs to do AVR, but certain strategies.

**Strengths:**

1. The problem of Abstract visual reasoning (AVR) is a popular problem and has gained great attention lately. The perspective of Bongard Problems is novel and meaningful.
2. The experiments of this paper are comprehensive on various MLLMs and 4 datasets (one of them is newly collected by this paper).
3. The dataset is already open-sourced, which may contribute to the research community in the future.

**Weaknesses:**

1. The contribution of the proposed Bongard-RWR dataset is not clear enough, although the authors briefly mentioned in Line 311-312. The previous datasets already covered both synthetic and real-world settings. How do the proposed datasets differentiate from these datasets? It is unclear why the authors find previous datasets insufficient or lacking in meaningfulness due to differences in concepts.
2. While the paper evaluates current MLLMs, it fails to offer a clear direction for enhancing their AVR capabilities or to present any new insights beyond highlighting general limitations. The results of different strategies might only reflect the training data (instruction tuning data) for these models.
3. Since the newly gathered dataset is considered as a main contribution, there are no clear statistics of this dataset provided.

**Questions:**

- Can you provide the rationale behind the proposed strategies in the experiments? Does it relate to the theory of AVR from a cognitive science perspective? Can you summarize the high-level idea of the design?
- To further understand the limitations, do you think the same model with different sizes (LLaVA 1.6 7b/13b/70b) will highlight some observations? I think it can reveal how scaling law will help with this problem.
- Can you list the key reasons why MLLMs fail in BPs? This should be provided in the conclusion section. You need to expand "broader limitations in their reasoning abilities" in Line 537-538.

---

> ### Author Response · Authors · 2024-11-23
> **Response to the Reviewer VpcT**
>
> > The contribution of the proposed Bongard-RWR dataset is not clear enough, [...]
>
> In preliminary experiments, we observed that while MLLMs perform well on Bongard HOI and Bongard-OpenWorld, they struggle with synthetic BPs. For instance, compare the best results in Images to Sides in Fig. 4 on synthetic BPs (75) to Bongard HOI (99) and Bongard-OpenWorld (96), or the max score in Table 1 on synthetic BPs (21) to Bongard HOI (45) and Bongard-OpenWorld (57). This discrepancy prompted us to investigate the reasons behind these differences.
>
> We identified two key factors that distinguish synthetic BPs from Bongard HOI and Bongard-OpenWorld. **Firstly, synthetic BPs feature simple 2D geometric shapes, while Bongard HOI and Bongard-OpenWorld consist of real-world images.** A potential explanation might be that the lack of training data on synthetic images could contribute to the lower performance on synthetic BPs. **Secondly, synthetic BPs focus on abstract concepts (e.g., “One line vs. Two lines” in Fig. 1a), while Bongard HOI and Bongard-OpenWorld involve concepts grounded in the real world (e.g., “A person jumping on a surfboard vs. Not a person jumping on a surfboard” in Fig. 1c).** This led us to hypothesize that MLLMs may find abstract concepts harder to recognize, especially when instantiated in various forms (see “$D^X_{i1}, D^X_{i2}, D^X_{i3}$” in Section 4.1).
>
> This motivated us to create the Bongard-RWR dataset, which bridges the gap between these two domains by representing abstract concepts with real-world images. As shown in Section 5, Bongard-RWR presents a major challenge to contemporary MLLMs, even though it relies on real-world images. This indicates that abstract concepts are inherently more difficult for models to recognize than concepts grounded in the real world. We believe that analyzing model performance across synthetic and real-world concepts opens a promising avenue for future work to fully explore this hypothesis. We have outlined ideas for extending this line of research in the newly added paragraph “Cross-domain analysis of MLLM perception” in Appendix A.
>
> > While the paper evaluates current MLLMs, it fails to offer a clear direction for enhancing their AVR capabilities or to present any new insights beyond highlighting general limitations. [...]
>
> Thank you for this comment. To better capture the current model limitations, we have restructured the presentation of results in Section 5. In particular, Fig. 4 (top) highlights a potential bias in MLLMs agreeing or disagreeing with presented concepts, which emphasizes the importance of considering natural language generation setups for evaluating MLLM abstract reasoning capabilities. Fig. 4 (bottom) illustrates that while MLLMs perform well in tasks involving real-world concepts (e.g., Bongard HOI and Bongard-OpenWorld), they struggle with abstract concepts (e.g., synthetic BPs and Bongard-RWR). We further show that MLLMs do not benefit from human-like approaches to solving BPs (see “Contrastive reasoning”), struggle to effectively utilize the context window (see “Iterative reasoning” and Descriptive-iterative in Fig. 5), and require further work to consistently integrate text and vision modalities at the answer generation step (see “Multimodal answer generation” and Descriptive-direct in Fig. 5).
>
> We believe that these insights provide clear recommendations for utilizing contemporary MLLMs to solve AVR tasks, and more generally, multi-image reasoning tasks. Specifically, the **Descriptive** strategy leads to the strongest performance, particularly when combined with its **-direct** variant in models that present stronger capabilities in integrating text and vision modalities, such as GPT-4o.
>
> Another direction for enhancing AVR capabilities emerges from the scaling law experiment discussed below in response to another comment. In summary, these results indicate that simply scaling model size may be insufficient to achieve stronger abstract reasoning capabilities. Future efforts are required to explicitly address this aspect, e.g., by incorporating AVR datasets into model training.
>
> In the revision we discuss additional ideas for future work in the newly introduced Appendix A.

---

> ### Author Response · Authors · 2024-11-23
> **Response to the Reviewer VpcT**
>
> > Since the newly gathered dataset is considered as a main contribution, there are no clear statistics of this dataset provided.
>
> Each problem instance in Bongard-RWR consists of 12 images (6 per side) and a text label, which limits the scope of traditional statistical summaries of the dataset itself. Instead, our analysis focused on evaluating the performance of different models and solution strategies, as detailed in Section 4. For instance, we noted several qualitative findings such as: *“This suggests that correctly identifying concepts expressed in Bongard-RWR likely requires more advanced reasoning abilities, even in the relatively simpler binary classification setting.“* or *“[...] the weak results on Bongard-RWR suggest that the discrepancy is more related to the specific underlying concepts than the visual domain as such [...]”*.
>
> We provide further information about Bongard-RWR in the supplementary materials:
> 1. Appendix F, Fig. 13 outlines the structure of the dataset based on the approach taken to construct each problem instance.
> 1. Appendix F, Fig. 14 compares selected samples from Bongard-RWR to their synthetic counterparts.
> 1. Appendix G contains detailed performance metrics across models and strategies, with Bongard-RWR results presented in Figs. 21 and 22.
> 1. Appendix H directly compares model performance on Bongard-RWR to synthetic BPs (Bongard, 1970).
>
> Additionally, in response to the Reviewers’ feedback, we have conducted a study involving human participants to relate model performance against a human benchmark. The study involved 30 human participants, as detailed in the newly introduced Appendix B.
>
> As shown in Fig. 7, humans solved an average of 39.2 problems, achieving 65% accuracy. Performance varied across participants, with the number of correctly solved problems ranging from 23 to 59, demonstrating that dedicated individuals are capable of solving almost all problems, and that in principle all 60 problems are solvable. Notably, the lowest number of problems solved by a human participant (23) exceeded the number of problems solved by all models in total (22, see “Solved by any model” in Fig. 22). In addition, Fig. 8 illustrates variability in problem difficulty: 22 problems were solved by at least 25 respondents, while 10 were solved by fewer than 10 test-takers. Notably, several problems easily solved by humans (e.g., 91, 92, 95, and 96) were not solved by any model, highlighting the need for further advances in this area.
>
> We hope that the above multi-faceted analysis offers a comprehensive view of the introduced dataset. If there are any particular statistics the Reviewer would find valuable that are missing in the paper, we would greatly appreciate further guidance.

---

> ### Author Response · Authors · 2024-11-23
> **Response to the Reviewer VpcT**
>
> > Can you provide the rationale behind the proposed strategies in the experiments? Does it relate to the theory of AVR from a cognitive science perspective? Can you summarize the high-level idea of the design?
>
> We chose these strategies to systematically explore and evaluate the capabilities and limitations of MLLMs in solving BPs, drawing inspiration from: (1) specificity of the task, (2) MLLM training setups, and (3) human problem-solving approaches.
>
> We began with the **Direct** strategy, the simplest and most intuitive approach, in which the model directly formulates an answer based on the provided image. It serves as a baseline to understand how MLLMs handle the task without any explicit intermediate reasoning steps. As discussed in Section 5 (“Generative capabilities in the Direct setting”), models overally underperform in this setting, which raised the need for formulating task-specific strategies to explore the limits of model abstract reasoning performance.
>
> To this end, we explored a task decomposition approach, where the reasoning process is broken down into smaller subtasks, whose solutions are later combined to solve the main task.
>
> The **Descriptive** strategy focuses on generating descriptions of individual images. This approach was inspired by the abundant amount of (image, text) pairs found in MLLM training datasets, which we expected would enable models to generate accurate image descriptions that facilitate abstract reasoning. As shown in Section 5 (“Independent image description”) the Descriptive strategy is generally preferred to other options, signifying the importance of selecting solution strategies that align with model training methods.
>
> Conversely, the **Contrastive** strategy focuses on identifying differences between image pairs composed of images from the opposite sides of the problem. Inspired by psychological insights into human problem-solving, particularly the role of contrast in highlighting differences (Nussli et al., 2009), we wanted to explore if MLLMs can benefit from such human-based strategies. As discussed in Section 5 (“Contrastive reasoning”), this strategy leads to worse results than Descriptive, highlighting clear differences in human and MLLM approaches to solving abstract reasoning tasks.
>
> Next, we considered **-iterative** variants of the **Descriptive** and **Contrastive** strategies. This design was motivated by the way humans iteratively refine their understanding as they incorporate new information. In addition, it compensates for the lack of memory in MLLMs by including past questions and responses within the reasoning context. As presented in Section 5 (“Iterative reasoning”), the results indicate that contemporary models struggle to effectively leverage additional information from the context window, raising the need for improving multi-step reasoning capabilities of MLLMs.
>
> Encouraged by the relatively strong performance of task decomposition methods compared to the Direct strategy, we hypothesized that combining these approaches could further enhance the abstract reasoning capabilities of MLLMs. To this end, we introduced **-direct** variants of the **Descriptive** and **Contrastive** strategies, allowing to cross-check individual observations against the entire matrix at the final answer generation step. However, the experiments presented in Section 5 (“Multimodal answer generation”) indicate that MLLMs face challenges in fully utilizing this additional multi-modal context to improve their reasoning performance.

---

> ### Author Response · Authors · 2024-11-23
> **Response to the Reviewer VpcT**
>
> > To further understand the limitations, do you think the same model with different sizes (LLaVA 1.6 7b/13b/70b) will highlight some observations? I think it can reveal how scaling law will help with this problem.
>
> Thank you for this excellent suggestion. We conducted an experiment to investigate how scaling the number of model parameters impacts AVR capabilities. To this end, we explored diverse model sizes including proprietary and open-access models. Firstly, we evaluated GPT-4o mini and Gemini 1.5 Flash, which are smaller alternatives of the already used GPT-4o and Gemini 1.5 Pro models, resp. Secondly, we employed several larger variants from the InternVL2 and LLaVA-NeXT model families. Overall, the set of considered models includes InternVL2-8B, InternVL2-26B, InternVL2-40B, InternVL2-Llama3-76B, LLaVA-v1.6 Vicuna-13B, LLaVA-v1.6 34B, LLaVA-NeXT 72B, and LLaVA-NeXT 110B. We conducted experiments on all 4 datasets using two solution strategies, including **Direct**, which is an intuitive baseline, and **Descriptive**, identified as the most effective strategy in the main experiments.
>
> Results and detailed analysis are presented in the newly introduced Appendix D. In summary, model scaling yields consistent improvements in AVR performance, especially for open-access models. Nonetheless, proprietary models exhibit strong performance even at smaller sizes, with GPT-4o mini performing at least as good as the best variant across open-access models in all but one cases.
>
> While these results show that model scaling can be indeed helpful in boosting AVR capabilities, they also suggest that large model size is not critical for strong performance in these tasks. Consequently, these results suggest that simply scaling the model size may be insufficient to achieve stronger abstract reasoning capabilities. It is highly relevant to investigate this aspect in future research, e.g., by incorporating AVR datasets into model training. Once again, we would like to thank you for this insightful comment.
>
> > Can you list the key reasons why MLLMs fail in BPs? This should be provided in the conclusion section. You need to expand "broader limitations in their reasoning abilities" in Line 537-538.
>
> Thank you for this suggestion. In response, we have restructured Section 6 to expand “broader limitations in their reasoning abilities” to specific insights drawn from our experiments.
>
> Furthermore, our evaluation revealed that models may fail to provide precise answers, especially when compared to human performance. For example, Fig. 12b highlights a case in which GPT-4o generated an almost correct concept (*“[...] Right: All images feature women in non-wedding attire, wearing dresses or suits of various colors other than white.”*), but overlooked a conflicting detail (one image featuring a woman in a white suit). This emphasizes the nuanced nature of BPs, where attention to details is critical for correctly recognizing the underlying concept. Developing robust MLLMs that consistently solve such tasks remains an open problem.

---

> ### Comment · Reviewer_VpcT · 2024-11-28
> **Response to the Rebuttal**
>
> Dear authors,
>
> I would like to thank you for your efforts to show the potential direction of this field and the statistics of the dataset. However, I am still not satisfied with the contribution part. The reason I am reiterating this aspect is that the dataset is the major contribution of this paper, thus it is critical to show what is new compared with previous datasets. It appears that the motivation is just bridging the Bongard problem in two domains (synthetic BPs and realistic BPs) by adapting them into another retrieved realistic domain, which I would regard as an augmentation technique to enrich the realistic data for the Bongard problems, as an increment to Bongard HOI and Bongard-OpenWorld.
>
> Additionally, the sample size of 100 raises methodological concerns. Given the potential for various sources of bias, this limited sample may not provide sufficient statistical power to draw robust conclusions. I would encourage the authors to provide a more detailed justification for why this sample size is adequate for the research objectives.
>
> Best,
> Reviewer VpcT

---

> > ### Author Response · Authors · 2024-11-29
> > **Response to Reviewer VpcT**
> >
> > We appreciate the reviewer’s comments and concerns about the contribution of Bongard-RWR and its sample size. While we acknowledge that a larger dataset would provide greater statistical power, our approach was shaped by two key factors.
> >
> > Firstly, Bongard-RWR was designed with significant manual effort, as detailed in Section 4.1. Of the 100 synthetic BPs considered, only 12 were automatically represented in the real-world domain using Algorithm 1. The remaining problems required manual construction, including the translation of abstract concepts to the real-world domain, image selection, and, in some cases,  photographing manually constructed scenes. This approach ensures that the dataset captures fine-grained abstract concepts, such as “Ends of the curve are parallel vs. Ends of the curve are perpendicular” or “Extensions of segments cross at one point vs. Extensions of segments do not cross at one point”, which are challenging to represent with real-world images. In contrast, datasets like Bongard HOI and Bongard-OpenWorld, which were generated automatically using existing data sources, focus on coarse-grained concepts such as “A person jumping on a surfboard. vs. Not a person jumping on a surfboard”. While this automated process enables larger sample sizes, it limits the representation of fine-grained abstract concepts.
> >
> > Secondly, the cost of MLLM inference needs to be accounted for. As detailed in Section 3.2, we employ an MLLM voting committee to evaluate generated answers, which comprises 4 proprietary models. The total number of inferences scales with the number of datasets, problem instances per dataset, models evaluated, and generation strategies employed. Scaling the experiments to much larger datasets is currently infeasible within our research budget.
> >
> > We hope that future advancements in optimizing MLLM inference costs will enable exploration of larger datasets. Until then, we believe that Bongard-RWR should provide a useful supplement to the existing real-world Bongard datasets (Bongard HOI and Bongard-OpenWorld) enabling testing of the abstract reasoning capabilities of machine learning models against detailed and nuanced aspects of the real-world scenes. We hope that the topic may attract other researchers to join this research path and contribute to the proposed version of the Bongard-RWR – the dataset will be made freely available for research purposes.

---

> > > ### Comment · Reviewer_VpcT · 2024-12-02
> > >
> > > Thank you for your response. However, it does not **directly** address my concerns regarding the dataset’s motivation and size. While I acknowledge the manual effort involved and the challenges of capturing fine-grained abstract concepts, the justification provided does not sufficiently differentiate Bongard-RWR from existing datasets like Bongard HOI and Bongard-OpenWorld, making it appear more like an incremental augmentation. Additionally, the small sample size of 100 remains a significant concern, as the justification does not convincingly demonstrate its adequacy for achieving the research objectives. Considering these issues alongside the feedback from other reviewers, I find it difficult to support acceptance of this paper.

---

> > > > ### Author Response · Authors · 2024-12-03
> > > > **Response to Reviewer VpcT**
> > > >
> > > > Dear Reviewer, thank you for your comments.
> > > >
> > > > We believe that Bongard-RWR is not merely an incremental augmentation of Bongard HOI and Bongard-OpenWorld, as we use a fundamentally different methodology to construct the dataset. While BPs in Bongard HOI and Bongard-OpenWorld are sampled automatically from a larger image dataset to represent concepts grounded in the real-world, we construct each BP manually, or in certain cases rely on a semi-automated process followed by manual review of the obtained matrices. Most importantly, **Bongard-RWR focuses on concepts not covered in Bongard HOI and Bongard-OpenWorld, making it orthogonal to these prior works**.
> > > >
> > > > The sample size of Bongard-RWR is indeed smaller than that of the remaining datasets, however, **the matrices qualitatively differ from existing datasets, by representing abstract concepts with real-world images**. We show that the introduced BPs pose a significant challenge for the contemporary methods, which we view as a novel and valuable direction to explore in the future work.

---

### Official Review · Reviewer_YYwM · 2024-11-08

**Soundness:** 2
**Presentation:** 3
**Contribution:** 3
**Rating:** 3
**Confidence:** 4

**Summary:**

This paper focuses on exploring one limitation of MLLM in visual understanding: there is still a big gap between MLLMs and humans in the IQ test / AVR test from psychometrics. They choose Bongard Problems to conduct the case study. Firstly, they build a new benchmark with some Bongard Problem samples. Then, they compare some SOTA MLLMs and VLMs with this multi-image reasoning task. The results suggest that the weak performance of MLLMs on Bongard Problems is not due to the domain specificity, but rather comes from their multi-image reasoning capability.

**Strengths:**

1. This is an important problem for MLLM nowadays. I like the motivation of your paper. As we know, Human IQ test has a long history in psychometrics and only a few of works try to explore the different between MLLM and human in this topic [1].

2. They have open-sourced their dataset containing 100 samples. I have a quick review of the dataset repo. The data structures and descriptions are easy to follow.

3. Compare with most of visual reasoning benchmark papers, this paper use less data but it has a solid cognition background. I happen to agree with some viewpoints of the authors, which are also highlighted by Google DeepMind recently.

References:
[1] Galatzer-Levy, Isaac R., et al. "The Cognitive Capabilities of Generative AI: A Comparative Analysis with Human Benchmarks." arXiv preprint arXiv:2410.07391 (2024).

**Weaknesses:**

1. It is worth to highlight that BP is only a subtask of AVR test. Compared to RAVEN, BP is not widely used by human's IQ test recently. I think you should discuss this limitation and consider exploring a board scope of AVR tasks in the future work.

2. The Evaluation settings in Figure 3 is unclear. It seems that most of the settings proposed by you can only use to test the closed-sourced models like Claude 3.5 and GPT-4o. The multi-image reasoning capbility of most of open-sourced MLLMs are not eligible in these tasks.

3. For a better understanding, the user study between human and MLLM is also important for this task.

4. Lacking an in-detailed analysis of the 100 samples in your datatset, e.g. the difficulty of each image pair.

**Questions:**

1. What is your insight on these proposed generation strategies in Figure 2? (Why do you choose these tasks?)

2. There are also some works from the observation in psychological experiments like MindSet [1], which could be the data sources for BP tasks or AVR tasks.

3. As your works show some similar observations to previous benchmark works like VCog-Bench, and François Chollet's book "On the measure of intelligence, you can put some further discussions in the main body of your paper.

Post Rebuttal:

I take a quick view of the authors' reply and other reviews' score, I decide to decrease my final score to 4. Though the authors choose a good topic as motivations, the main pain points (over claiming, lacking data, lacking detailed definition) are difficult to solve. This, we all agree this paper doesn't match the bar of ICLR.

---

> ### Author Response · Authors · 2024-11-23
> **Response to the Reviewer YYwM**
>
> > [...] As we know, Human IQ test has a long history in psychometrics and only a few of works try to explore the different between MLLM and human in this topic [1].
>
> Thank you for pointing out this highly relevant study. As it was published after the ICLR submission deadline, we were unable to include it in the initial version of our submission. However, based on your valuable suggestion, we have now incorporated a discussion of this work in the newly added Appendix A, within the paragraph “Incorporating proposed strategies to enhance abstract reasoning abilities”.
>
> > It is worth to highlight that BP is only a subtask of AVR test. [...]
>
> We focused our study on BPs because they fundamentally require solvers to articulate answers in natural language, making them a valuable testbed for evaluating the reasoning capabilities of MLLMs. In contrast, tasks like Raven’s Progressive Matrices pose a discriminative challenge, which can make it difficult to distinguish whether a model’s solution is a result of true understanding or educated guess. Nevertheless, we agree that BPs represent only a subset of AVR tasks and that exploring a broader range of challenges is essential for advancing the field. To address this limitation, we have added a discussion in the newly included paragraph “Going Beyond Bongard Problems” in Appendix A.
>
> > [...] The multi-image reasoning capbility of most of open-sourced MLLMs are not eligible in these tasks.
>
> We specifically selected open-source models that support multi-image reasoning, as confirmed in their documentation:
> 1. https://huggingface.co/mistralai/Pixtral-12B-2409 “You can also pass multiple images per message [...]“
> 1. https://huggingface.co/OpenGVLab/InternVL2-8B “InternVL 2.0 is trained with an 8k context window and utilizes training data consisting of long texts, multiple images, and videos, [...]”
> 1. https://huggingface.co/microsoft/Phi-3.5-vision-instruct “The model provides uses for general purpose AI systems and applications with visual and text input capabilities which require: [...] 6. Multiple image comparison 7. Multi-image or video clip summarization [...]”
> 1. https://huggingface.co/docs/transformers/main/en/model_doc/llava_next#multi-image-inference “LLaVa-Next can perform inference with multiple images as input, [...]“
>
> Out of the 10 solution strategies employed in our work, 4 (Images to Sides, Contrastive, Contrastive-iterative, and Contrastive-direct) explicitly involve processing multiple images in a single forward pass. However, we recognize that multi-image training data is significantly less available compared to standard (image, text) pairs, which may partially explain the weaker performance of the Contrastive strategies compared to Descriptive ones. We believe that advancing multi-image training datasets is a critical step towards improving model performance in this regime.
>
> > For a better understanding, the user study between human and MLLM is also important for this task.
>
> Thank you for your valuable suggestion. To evaluate the difficulty and validity of the proposed dataset, during the rebuttal period, we conducted a study with 30 human participants, as detailed in the newly introduced Appendix B. As shown in Fig. 7, humans solved an average of 39.2 problems, achieving 65% accuracy. Performance varied across participants, with the number of correctly solved problems ranging from 23 to 59, demonstrating that dedicated individuals are capable of solving almost all problems, and that in principle all 60 problems are solvable. Notably, the lowest number of problems solved by a human participant (23) exceeded the number of problems solved by all models in total (22, see “Solved by any model” in Fig. 22). In addition, Fig. 8 illustrates variability in problem difficulty: 22 problems were solved by at least 25 respondents, while 10 were solved by fewer than 10 test-takers. Notably, several problems easily solved by humans (e.g., 91, 92, 95, and 96) were not solved by any model, highlighting the need for further advances in this area.
>
> These findings highlight significant gaps between human and model-based performance on this dataset, demonstrating the need for further development of MLLM AVR abilities.
>
> > Lacking an in-detailed analysis of the 100 samples in your datatset, e.g. the difficulty of each image pair.
>
> To address this concern, we conducted an analysis based on participant feedback from our human study. Participants rated the overall difficulty of all problems on a scale from 1 to 10, resulting in an average difficulty of 7.6, indicating that the dataset is perceived as quite challenging. In addition, Fig. 8 illustrates variability in problem difficulty: 22 problems were solved by at least 25 respondents, while 10 were solved by fewer than 10. Notably, several problems easily solved by humans (e.g., 91, 92, 95, and 96) were not solved by any model, highlighting the need for further advances in this area.

---

> ### Author Response · Authors · 2024-11-23
> **Response to the Reviewer YYwM**
>
> > What is your insight on these proposed generation strategies in Figure 2? (Why do you choose these tasks?)
>
> We chose these strategies to systematically explore and evaluate the capabilities and limitations of MLLMs in solving BPs, drawing inspiration from: (1) specificity of the task, (2) MLLM training setups, and (3) human problem-solving approaches.
>
> We began with the **Direct** strategy, the simplest and most intuitive approach, in which the model directly formulates an answer based on the provided image. It serves as a baseline to understand how MLLMs handle the task without any explicit intermediate reasoning steps. As discussed in Section 5 (“Generative capabilities in the Direct setting”), models overally underperform in this setting, which raised the need for formulating task-specific strategies to explore the limits of model abstract reasoning performance.
>
> To this end, we explored a task decomposition approach, where the reasoning process is broken down into smaller subtasks, whose solutions are later combined to solve the main task.
>
> The **Descriptive** strategy focuses on generating descriptions of individual images. This approach was inspired by the abundant amount of (image, text) pairs found in MLLM training datasets, which we expected would enable models to generate accurate image descriptions that facilitate abstract reasoning. As shown in Section 5 (“Independent image description”) the Descriptive strategy is generally preferred to other options, signifying the importance of selecting solution strategies that align with model training methods.
>
> Conversely, the **Contrastive** strategy focuses on identifying differences between image pairs composed of images from the opposite sides of the problem. Inspired by psychological insights into human problem-solving, particularly the role of contrast in highlighting differences (Nussli et al., 2009), we wanted to explore if MLLMs can benefit from such human-based strategies. As discussed in Section 5 (“Contrastive reasoning”), this strategy leads to worse results than Descriptive, highlighting clear differences in human and MLLM approaches to solving abstract reasoning tasks.
>
> Next, we considered **-iterative** variants of the **Descriptive** and **Contrastive** strategies. This design was motivated by the way humans iteratively refine their understanding as they incorporate new information. In addition, it compensates for the lack of memory in MLLMs by including past questions and responses within the reasoning context. As presented in Section 5 (“Iterative reasoning”), the results indicate that contemporary models struggle to effectively leverage additional information from the context window, raising the need for improving multi-step reasoning capabilities of MLLMs.
>
> Encouraged by the relatively strong performance of task decomposition methods compared to the Direct strategy, we hypothesized that combining these approaches could further enhance the abstract reasoning capabilities of MLLMs. To this end, we introduced **-direct** variants of the **Descriptive** and **Contrastive** strategies, allowing to cross-check individual observations against the entire matrix at the final answer generation step. However, the experiments presented in Section 5 (“Multimodal answer generation”) indicate that MLLMs face challenges in fully utilizing this additional multi-modal context to improve their reasoning performance.
>
> > There are also some works from the observation in psychological experiments like MindSet [1], which could be the data sources for BP tasks or AVR tasks.
>
> Thank you for bringing this relevant reference to our attention. We agree that evaluating MLLMs using the proposed strategies on the MindSet: Vision toolbox represents a valuable direction for future research. Such an investigation could provide deeper insights into object perception and concept identification in MLLMs, further clarifying their alignment with human perceptual reasoning. We have included a discussion of this idea in Appendix A under the paragraph “Fine-grained analysis of MLLM perception”.
>
> > As your works show some similar observations to previous benchmark works like VCog-Bench, and François Chollet's book [...]
>
> Thank you for pointing out the relevance of these works. We agree that they are closely related to our study and have already cited them in Section 2: *“Recent research concerning the evaluation of abstract reasoning skills of LLMs concentrates around the Abstraction and Reasoning Corpus (ARC) task (Chollet, 2019).”* and *“Cao et al. (2024) proposed a suite of AVR tasks to compare MLLM and human performance.”* Given the breadth of our study, it was challenging to expand the discussion of these works within the main body of the paper. However, we have added a detailed discussion in the newly introduced paragraph “Going Beyond Bongard Problems” in Appendix A to address these works more comprehensively.

---

> ### Comment · Reviewer_YYwM · 2024-11-26
> **Response and further concerns**
>
> Dear Authors of ICLR 2025 Submission #6466,
>
> Thank you for your response and for updating your manuscript. I have carefully reviewed your revised version and the replies from the other reviewers. I would like to raise some new concerns, and I am considering adjusting my score based on these points.
>
> **Human Study**
>
> I am happy you follow Reviewer VQyh's and my suggestions to conduct the human study, but the current version is not satisfied.
>
> The methodology of your cognition and psychometrics human study raises several concerns regarding its rigor. For a robust human study, it is essential to first obtain IRB approval from your institution. Additionally, participants should be recruited based on clear criteria (e.g., data selection process, age range, gender distribution, IQ levels). Moreover, your results seem to contradict your previous statements. You mention that "Humans often rely on direct comparisons between image panels from different categories to highlight differences, whereas the tested methods perform better when making comparisons on text-based image descriptions..." However, your human study shows that participants could solve only 65% of the problems in your dataset, and it remains unclear how humans approach solving the Bongard problems in your dataset.
>
> You have not provided evidence to demonstrate whether the suboptimal human performance is due to your data design or the inherent difficulty of the problems. This issue appears to be more related to HCI and psychometrics, and it necessitates a rigorously designed experiment to investigate it thoroughly.
>
> **Insights from Your Rebuttal**
>
> I have concerns regarding the insights you proposed in your rebuttal. You state that MLLMs still face challenges even after applying all your improved strategies. Identifying challenges faced by MLLMs is not novel in itself. For instance, Kian's paper, "The Curious Case of Nonverbal Abstract Reasoning with Multi-modal Large Language Models," explores similar issues, and there is significant overlap with your insights.
>
> To strengthen your statement, I suggest re-evaluating the essence of solving the Bongard Problem based on Kian's findings, including adding experiments for few-shot learning, chain-of-thought, training-testing experiments, RLHF finetuning. This could provide a deeper understanding and highlight the unique contributions of your work.
>
> **Comparison with Existing Datasets**
>
> Reviewer VpcT raised an important question that I had not initially noticed. You state, "Note, however, that a direct performance comparison on synthetic Bongard datasets versus real-world Bongard HOI and Bongard-OpenWorld datasets is not meaningful, as these datasets depict different concepts." This prompted me to examine the data from Bongard HOI and Bongard-OpenWorld.
>
> From my perspective, there does not seem to be a significant difference between their datasets and yours. Your logic appears to be that because MLLMs perform well on Bongard HOI and Bongard-OpenWorld, these datasets are less meaningful, whereas the poor performance on your dataset proves yours value. This reasoning is not rigorous and raises several concerns. For example, readers might suspect that your data were selectively chosen to highlight the weaknesses of MLLMs, potentially contradicting the definition of Bongard Problems, which could undermine the validity of your conclusions.
>
> I recommend that you address the concerns raised by Reviewer VpcT thoroughly. In my view, this is a core issue with your paper, and your current response does not adequately resolve it. I am open to further discuss these problems with you,  Reviewer VQyh, and Reviewer VpcT during the rebuttal. Let's wait for other reviewers' reply.
>
>
> Reviewer YYwM

---

> > ### Author Response · Authors · 2024-11-26
> > **Response to Reviewer YYwM regarding further concerns (1 / 2)**
> >
> > Dear Reviewer YYwM, thank you for reviewing the revised paper and our response to the Reviewers.
> >
> > > Human study
> >
> > We fully agree that human studies focused on understanding human cognition and psychometrics necessitate additional methodological rigor and clear selection criteria. However, given the limited time available during the rebuttal period, it was not feasible to conduct a more comprehensive study on a larger scale. It is worth noting that while the details of the human study on Bongard-OpenWorld are not reported, Bongard-HOI employed 35 participants, which is comparable to the scale of our study conducted within this constrained timeline.
> >
> > Consecutively, our human study was not intended to analyze human cognition in depth or investigate the specific strategies humans employ when solving BPs. Instead, our primary objective was to provide a reference point for human performance, contextualizing the results of MLLMs and demonstrating usability and solvability of the dataset. The average human accuracy of 65% (with the best individual performance reaching 98.3%) significantly exceeds the best performance achieved by MLLMs, where the highest result (Claude 3.5 Sonnet using the Descriptive strategy) was 22%, solving only 13 out of 60 problems. This comparison shows a substantial gap between human and MLLM performance, highlighting the challenges posed by Bongard-RWR. While we acknowledge that our study is insufficient to draw detailed insights regarding human cognition, we believe it effectively serves its intended purpose of establishing a reference for assessing MLLM results.
> >
> > Our observation regarding human approaches to solving BPs is grounded in prior research, see “Contrastive reasoning” in Section 5: *“Humans often rely on direct comparisons between image panels from different categories to highlight differences __(Nussli et al., 2009)__”*. Investigating the specific cognitive processes or strategies humans use to solve Bongard-RWR was beyond the scope of our work. We recognize this as an interesting direction for future research and appreciate the Reviewer’s suggestion to approach this in greater depth.
> >
> > > Insights from Your Rebuttal
> >
> > Thank you for bringing this highly relevant work to our attention. We were unaware of (Kian et al., 2024) during the paper submission, as it was accepted for publication on 10th July, 2024 ([OpenReview link](https://openreview.net/forum?id=eDWcNqiQWW)). According to the ICLR FAQ, such works are considered concurrent to ICLR submissions. We will thoroughly investigate this paper and incorporate its findings in our future works.
> >
> > While we do not explore all the suggested topics in the same depth, our experiments already align with some of the methods highlighted in (Kian et al., 2024). For example, we utilize few-shot learning in Prompts 6 and 7, where in-context examples of expected solutions are provided. In addition, the Descriptive and Contrastive strategies, together with their -direct and -iterative variants, can be considered as implementations of chain-of-thought reasoning, where partial descriptions of BP images form subsequent observations summarized at the final answer generation step. While we do not explicitly perform model training or RLHF fine-tuning, we recognize the value of these approaches and agree they form promising directions for future exploration.
> >
> > Nevertheless, in this paper we have aimed to provide a broad contribution by considering 3 datasets from the literature, introducing a new dataset, evaluating 8 models (both proprietary and open-access), exploring 7 natural language generation strategies and 3 binary classification tasks. Moreover, we conducted a scaling law experiment and a human study to contextualize MLLM performance. While we acknowledge that incorporating suggested experiments would provide additional insights, we hope the breadth and depth of our current contributions may be considered valuable on their own.

---

> > ### Author Response · Authors · 2024-11-26
> > **Response to Reviewer YYwM regarding further concerns (2 / 2)**
> >
> > > Comparison with Existing Datasets
> >
> > We respectfully disagree with the notion that we consider MLLM results on Bongard HOI and Bongard-OpenWorld to be less meaningful. On the contrary, we believe these datasets play an essential role in evaluating model performance on concepts grounded in the real-world. The contributions of Bongard HOI and Bongard-OpenWorld are complementary to our work.
> >
> > The development of Bongard-RWR was motivated by observed discrepancies in model performance between synthetic BPs and real-world datasets including Bongard HOI and Bongard-OpenWorld, as outlined in our response to the Reviewer VpcT: *“In preliminary experiments, […] This discrepancy prompted us to investigate the reasons behind these differences.”* Through this investigation, we identified two main ideas differentiating synthetic BPs from Bongard HOI and Bongard-OpenWorld, as further explained in the mentioned response: *“We identified two key factors that distinguish synthetic BPs from Bongard HOI and Bongard-OpenWorld. [...]”*.
> >
> > Bongard-RWR was specifically designed to fill this gap, by introducing BPs comprising real-world images presenting abstract concepts. It is not intended to serve as a more meaningful dataset but rather to complement synthetic BPs, Bongard HOI and Bongard-OpenWorld.
> >
> > > I recommend that you address the concerns raised by Reviewer VpcT thoroughly.
> >
> > In our response to Reviewer VpcT, we provided a detailed explanation of the motivation behind Bongard-RWR, outlined key insights and recommendations from our experiments, extended the dataset statistics with data from the human study, explained the rationale for the introduced strategies, analyzed the influence of model scaling on abstract reasoning performance, and expanded the paper’s conclusions. If there are any specific aspects of our response that require further clarification, we would be happy to address them in more detail.

---

> > > ### Comment · Reviewer_YYwM · 2024-12-01
> > > **Response and further concerns**
> > >
> > > Thank you for your response. Regarding the first two questions, I understand that they might be challenging to address at this stage. However, for the last question, I would need a more detailed explanation. I have carefully reviewed your reply as well as Reviewer VpcT's follow-up, and it seems that the main issue remains unresolved. Specifically, the prior datasets already represent a significant contribution to this area, while your dataset appears to be a minor augmentation (like Reviewer VpcT proposed). However, the way your paper is written gives the impression that your contribution is overstated. Reviewer VpcT observed this problem and summaried into his/her/their comment.
> > >
> > > Based on Reviewer VpcT's comment, I want highlight another problem: the data selection process for your dataset remains unclear. As mentioned in your paper, the GPT-4o data generation process involves manual intervention at several steps. This raises concerns about how these manually selected images align with the design standards of Bongard problems. As I pointed out earlier, a potential risk is that readers might question whether your dataset was manually selected to emphasize the weaknesses of MLLMs, particularly in terms of the data are sampling from conditional probability distributions. This could contradict the foundational principles of Bongard problems and potentially undermine the validity of your conclusions.
> > >
> > > To effectively explore MLLM performance and facilitate meaningful comparisons between human and MLLM capabilities, it would be more appropriate to use a widely collected dataset in general domain (e.g., sampling natural and symbolic images in Bongard HOI and Bongard-OpenWorld, and other datasets) for Bongard problems and evaluate the models within that distribution. This contradicts your current conclusion.

---

> > > > ### Author Response · Authors · 2024-12-03
> > > > **Response to Reviewer YYwM**
> > > >
> > > > First of all, thank you for your thorough review of our paper and pointing out important directions to explore in the future work.
> > > >
> > > > > Specifically, the prior datasets already represent a significant contribution to this area, while your dataset appears to be a minor augmentation (like Reviewer VpcT proposed). However, the way your paper is written gives the impression that your contribution is overstated. Reviewer VpcT observed this problem and summaried into his/her/their comment.
> > > >
> > > > We acknowledge the contributions of Bongard HOI and Bongard-OpenWorld and aimed to make it clear in the paper that Bongard-RWR connects synthetic BPs with real-world BPs introduced in these two datasets, instead of trying to replace them. In particular, we motivate and summarize the dataset’s contribution as: *"To further examine the main difficulties faced by MLLMs in solving both types of BPs (synthetic and real world ones) we introduce a focused dataset of BPs (Bongard-RWR) comprising real-world images that represent concepts from synthetic BPs using real world images. Thanks to relying on the same abstract concepts as synthetic BPs, Bongard-RWR facilitates direct comparisons of the MLLMs performance in both domains."* and *"To delve deeper into the performance discrepancies between synthetic and real-world domains, we introduced Bongard- RWR, a new BP dataset designed to represent concepts from synthetic BPs via real-world images."*
> > > >
> > > > We also think that Bongard-RWR is not merely an augmentation of these two datasets, as we use a fundamentally different methodology to construct the dataset. While BPs in Bongard HOI and Bongard-OpenWorld are sampled automatically from a larger image dataset to represent concepts grounded in the real-world, we construct each BP manually, or in certain cases rely on a semi-automated process followed by manual review of the obtained matrices. Most importantly, Bongard-RWR focuses on concepts not covered in Bongard HOI and Bongard-OpenWorld, making it orthogonal to these prior works.
> > > >
> > > > Finally, we emphasize that Bongard-RWR is only one aspect of the broader contributions presented in our study, as detailed in our previous response: *"[...] in this paper we have aimed to provide a broad contribution [...]"*. We hope that the overall breadth and depth of our paper constitutes a valuable study on the reasoning limitations of MLLMs.
> > > >
> > > > > readers might question whether your dataset was manually selected to emphasize the weaknesses of MLLMs, particularly in terms of the data are sampling from conditional probability distributions. This could contradict the foundational principles of Bongard problems and potentially undermine the validity of your conclusions.
> > > >
> > > > We appreciate the concern and want to clarify that our dataset was not manually curated to emphasize the weaknesses of MLLMs. To construct matrices in Bongard-RWR, we focused on selecting diverse images that accurately reflect concepts used in synthetic BPs via real-world images. We did not take into account model, nor human, performance in solving these tasks. In fact, the original synthetic BPs were likewise designed manually and the set of BPs was further extended by individual contributors, which follows the approach taken to design Bongard-RWR.
> > > >
> > > > > To effectively explore MLLM performance and facilitate meaningful comparisons between human and MLLM capabilities, it would be more appropriate to use a widely collected dataset in general domain (e.g., sampling natural and symbolic images in Bongard HOI and Bongard-OpenWorld, and other datasets) for Bongard problems and evaluate the models within that distribution. This contradicts your current conclusion.
> > > >
> > > > The primary goal of our research was not to compare human and MLLM capabilities. Instead, we introduced a new dataset to investigate why MLLMs perform poorly on synthetic BPs compared to Bongard-HOI and Bongard-OpenWorld. A study focused solely on Bongard-HOI and Bongard-OpenWorld would be inconclusive, as these datasets do not capture the same concepts as synthetic BPs. The human study was conducted solely to serve as a general baseline. However, we agree that the robustness of the presented results could be further supported by a larger sample size. In the future work we will consider constructing additional instantiations of the selected matrices to ensure model performance is not heavily influenced by the individual selected images.

---

### Official Review · Reviewer_VQyh · 2024-11-10

**Soundness:** 3
**Presentation:** 3
**Contribution:** 2
**Rating:** 5
**Confidence:** 4

**Summary:**

The paper focuses on the capability of multimodal large language models (MLLM) to solve Bongard Problems (BP). The authors propose a new BP dataset, where the two sides of the images are distinguished by abstract visual concepts. The authors design a set of evaluation methods for MLLMs, including binary classifications and solving BP using different generation strategies. The results on the proposed BP dataset and three other BP datasets show that the abstract visual reasoning capabilities of MLLMs are limited in solving BP, especially in the proposed ones.

**Strengths:**

The paper provides a comprehensive evaluation of the capabilities of vision language models on Bongard Problems (BPs). The paper includes eight models and 10 metrics to evaluate them on 4 types of BPs, including one created by the authors. I believe it will be very helpful for future study of BPs.

**Weaknesses:**

- The presentation of the paper needs to be improved. Despite numerous settings, new generation strategies and new dataset, the authors choose to present the results in just one table. It is almost impossible to interpret the table and verify the claims made by the authors without checking back and forth. I suggest reviewer break down the table into small ones, where each of them provides a clear message to the readers. If that takes more space than the current version, some of the experiment results can be moved to appendix.
- The observation from the results is currently not very interesting. The major claim made by the authors is that the capability of the VLMs on BPs are limited. Not many intuitions behind are obtained, part of it is because of the poor presentation, and the other part is probably because the observations are not very consistent across different models. I suggest the authors dive deep into the details and conduct additional experiments if necessary. Overall, I believe the contribution of the paper can be improved.
- The new BP dataset created in the paper is quite difficult for the models. I believe a comprehensive human study is necessary in this case to ensure the validity of the dataset. It is great that the dataset is released, but after a quick scan of the dataset, I personally find it very hard to distinguish the left and the right images for some BPs. Therefore, I suggest the authors to report human performance on the tasks and report consistency across human annotators.


------
### Post-rebuttal comments:
- The authors added a human study of the new BP dataset and re-organized the experiment section, which are important improvements for the paper. Therefore, I have raised my score.
- I agree with other reviewers that the dataset and the analysis can be made more rigorous and comprehensive. Therefore, I believe this paper has not reached the bar of NeurIPS.

**Questions:**

It would be nice to show some human score for BPs. Is there any attempt toward that?

---

> ### Author Response · Authors · 2024-11-23
> **Response to the Reviewer VQyh**
>
> > The presentation of the paper needs to be improved. [...]
>
> Thank you for this suggestion. In response, we have restructured our presentation to improve clarity and accessibility. The detailed table and discussion of binary classification tasks have been moved to Appendix C. A concise summary of binary classification results is now provided in a single paragraph accompanied by the newly introduced Fig. 4. Additionally, we have introduced Table 1, which highlights the performance of Direct, Descriptive, and Contrastive strategies, and Fig. 5, which illustrates the impact of the -direct and -iterative variants of the Descriptive and Contrastive strategies. We hope this revised format simplifies interpretation of results and improves the clarity of the insights drawn from our experiments.
>
> > The observation from the results is currently not very interesting. [...]
>
> We hope that our response to the previous comment addresses the first part regarding poor presentation. In particular, we believe that Fig. 4 (top) highlights a potential bias in MLLMs agreeing or disagreeing with presented concepts, which emphasizes the importance of considering natural language generation setups for evaluating MLLM abstract reasoning capabilities. Fig. 4 (bottom) illustrates that while MLLMs perform well in tasks involving real-world concepts (e.g., Bongard HOI and Bongard-OpenWorld), they struggle with abstract concepts (e.g., synthetic BPs and Bongard-RWR). We further show that MLLMs do not benefit from human-like approaches to solving BPs (see “Contrastive reasoning”), struggle to effectively utilize the context window (see “Iterative reasoning” and Descriptive-iterative in Fig. 5), and require further work to consistently integrate text and vision modalities at the answer generation step (see “Multimodal answer generation” and Descriptive-direct in Fig. 5).
>
> We agree that some observations vary across different models, but we believe this is an expected outcome given the diversity of MLLMs in terms of the data mixtures used during pre-training, fine-tuning and learning from human feedback, the architectures of their text and vision encoders, and the approaches for combining multimodal embeddings. The unique nature of BPs may further contribute to variability in model performance, as these analogy-based reasoning tasks are largely absent in standard MLLM data sources, which primarily include visual question answering or image captioning tasks.
>
> To further investigate this variability, during the rebuttal time, we conducted an experiment exploring the impact of scaling model parameters on AVR capabilities. We explored diverse model sizes including both proprietary and open-access models. Firstly, we evaluated GPT-4o mini and Gemini 1.5 Flash, which are smaller alternatives of the already used GPT-4o and Gemini 1.5 Pro models, resp. Secondly, we employed several larger variants from the InternVL2 and LLaVA-NeXT model families. Overall, the set of considered models includes InternVL2-8B, InternVL2-26B, InternVL2-40B, InternVL2-Llama3-76B, LLaVA-v1.6 Vicuna-13B, LLaVA-v1.6 34B, LLaVA-NeXT 72B, and LLaVA-NeXT 110B. We conducted experiments on all 4 datasets using two solution strategies, including Direct, which is an intuitive baseline, and Descriptive, identified as the most effective strategy in the main experiments.
>
> Results and detailed analysis are presented in the newly introduced Appendix C. In summary, model scaling yields consistent improvements in AVR performance, especially for open-access models. Nonetheless, proprietary models exhibit strong performance even at smaller sizes, with GPT-4o mini performing at least as good as the best variant across open-access models in all but one cases.
>
> While these results show more consistent trends within a single model architecture, some inconsistencies remain, as certain smaller variants outperform their larger counterparts. This suggests that future efforts to improve MLLM abstract reasoning capabilities should devote larger focus to this aspect, e.g., by incorporating AVR datasets into model training.

---

> ### Author Response · Authors · 2024-11-23
> **Response to the Reviewer VQyh**
>
> > The new BP dataset created in the paper is quite difficult for the models. [...]
>
> Thank you for your valuable suggestion. To evaluate the difficulty and validity of the proposed dataset, during the rebuttal period, we conducted a study with 30 human participants, as detailed in the newly introduced Appendix B. As shown in Fig. 7, humans solved an average of 39.2 problems, achieving 65% accuracy. Performance varied across participants, with the number of correctly solved problems ranging from 23 to 59, demonstrating that dedicated individuals are capable of solving almost all problems, and that in principle all 60 problems are solvable. Notably, the lowest number of problems solved by a human participant (23) exceeded the number of problems solved by all models in total (22, see “Solved by any model” in Fig. 22). In addition, Fig. 8 illustrates variability in problem difficulty: 22 problems were solved by at least 25 respondents, while 10 were solved by fewer than 10 test-takers. Notably, several problems easily solved by humans (e.g., 91, 92, 95, and 96) were not solved by any model, highlighting the need for further advances in this area.
>
> > It would be nice to show some human score for BPs. Is there any attempt toward that?
>
> Human performance has also been evaluated for Bongard HOI and Bongard-OpenWorld, where participants achieved an average accuracy of 91% on both datasets. In comparison, the 65% accuracy observed in our study highlights that Bongard-RWR poses a greater challenge even for humans. Nonetheless, the high scores achieved by certain participants demonstrate that all tasks in Bongard-RWR are solvable with dedicated effort, making it a valuable and challenging testbed for advancing research in this area.

---

### Author Response · Authors · 2024-11-23
**Response to all Reviewers**

Dear Reviewers, we are grateful for your insightful comments and suggestions, which helped us to improve the paper. Following the received feedback, during the rebuttal time we improved the presentation of results in Section 5 by redesigning Table 1 and introducing Figs. 4 and 5. Additionally, we summarized key findings in Section 6, discussed limitations and opportunities for future work (Appendix A), conducted a human study including 30 participants to assess Bongard-RWR difficulty (Appendix B), moved the detailed discussion of binary classification tasks and the table with full results from the main paper to Appendix C, and performed a scaling law experiment (Appendix D). Labels of figures, tables, and sections mentioned in this response refer to the revised version of the manuscript. Specific comments are addressed in the individual responses to each Reviewer. We hope the provided responses adequately address the Reviewers' concerns.

---

### Meta-Review · Area_Chair_qeHQ · 2024-12-21

**Metareview:**

The AC acknowledges the authors’ efforts in exploring a potential direction for this field and providing dataset statistics. However, the paper's primary contribution, the dataset, lacks sufficient novelty, after careful checking and syncing with reviwers' commets, as it appears to primarily bridge synthetic and realistic Bongard problems through a data augmentation technique, building incrementally on existing datasets like Bongard HOI and Bongard-OpenWorld. Furthermore, the limited sample size of 100 raises methodological concerns regarding potential bias and insufficient statistical power, with no detailed justification provided for its adequacy. While the work has merit, these limitations lead to a reject recommendation for now.

**Additional Comments On Reviewer Discussion:**

Reviewers engaged in discussion and with further concerns.

---

### Decision · Program_Chairs · 2025-01-22

Reject